# Observation of current-induced bulk magnetization in elemental tellurium

Tetsuya Furukawa [1], Yuri Shimokawa[1], Kaya Kobayashi[2] & Tetsuaki Itou[1]

The magnetoelectric effect in bulk matter is of growing interest both fundamentally and technologically. Since the beginning of the century, the magnetoelectric effect has been studied intensively in multiferroic materials. However, magnetoelectric phenomena in materials without any (anti-)ferroic order remain almost unexplored. Here we show the observation of a new class of bulk magnetoelectric effect, by revisiting elemental trigonal tellurium. We demonstrate that elemental tellurium, which is a nonmagnetic semiconductor, exhibits current-induced magnetization. This effect is attributed to spin splitting of the bulk band owing to the lack of inversion symmetry in trigonal tellurium. This finding highlights magnetoelectricity in bulk matter driven by moving electrons without any (anti-)ferroic order. Notably, current-induced magnetization generates a magnetic field that is not circular around but is parallel to the applied current; thus, this phenomenon opens a new area of magnetic field generation beyond Ampere's law that may lead to industrial applications.

[1] Department of Applied Physics, Tokyo University of Science, 6-3-1 Niijyuku, Katsushika-ku, Tokyo 125-8585, Japan. [2] Research Institute for Interdisciplinary Science, Okayama University, 3-1-1 Tsushimanaka, Kita-ku, Okayama 700-8530, Japan. Correspondence and requests for materials should be addressed to T.F.(email: tetsuya.furukawa@rs.tus.ac.jp) or to T.I.(email: tetsuaki.itou@rs.tus.ac.jp)

The study of magnetoelectricity has been sparked by the recent development of multiferroic materials[1]. Such materials possess coexisting magnetic and electric order, enabling the control of electronic magnetization by the application of an electric field. However, the spin-orbit interaction, which causes moving electrons in solids to experience an effective magnetic field as a consequence of relativity, can provide an alternative way to realize magnetoelectricity in non-ferroic materials. This effective magnetic field plays a key role, particularly under conditions of inversion asymmetry, generating spin-split energy bands with oppositely spin-polarized states[2–5]. Such spin-split bands are widely known in the fields of surface and interface physics; the spin-orbit interaction and inversion asymmetry due to the confinement potential at a surface or interface produce so-called Bychkov-Rashba spin-split surface bands[4]. Thus, although the net spin polarization is exactly zero in equilibrium, an electric current causes uniform spin polarization owing to an imbalance between populations of up and down spins, which is called current-induced spin polarization[6–8]. This effect has been experimentally observed for surfaces[9], interfaces[10–14], and epilayers[15–17], which are not bulk systems. (In this study, the term bulk refers to an infinite system free of boundary effects including the local strain caused by a heterointerface.) To extend this phenomenon to bulk physics is an intriguing issue. An applied current can induce bulk electronic magnetization in non-centrosymmetric materials (to be precise gyrotropic[18] non-centrosymmetric materials), because the inversion asymmetry in

bulk crystals produces spin-split bulk bands. Elemental tellurium, which is a bulk crystal of a chiral semiconductor composed of heavy atoms, is an ideal playground for this issue. Indeed, it has been reported that applying an electric current to bulk tellurium causes the current induced optical activity[8, 19]. This is the first experimental result that captured a sign of the current-induced electronic magnetization in bulk tellirum[19], although careful discussions, such as about separation of this additional optical activity from the inherent natural optical activity, are needed. To firmly establish the current-induced bulk electronic magnetization in tellurium, it is required to detect it simply by a probe capable of sensitive and microscopic detection of local electronic magnetization.

Here we demonstrate the observation of current-induced electronic magnetization in bulk tellurium using nuclear magnetic resonance (NMR) measurements. We present the current-induced shift of a $^{125}$Te-NMR spectrum depending on the strength and the polarity of an applied electric current; this result indicates that an applied current induces bulk electronic magnetization. This finding provides evidence of magnetoelectricity in bulk matter without (anti-)ferroic order.

## Results

**Crystal and electronic structures of elemental tellurium.** Elemental tellurium forms a trigonal crystal structure [$P3_121$ ($D_3^4$) or $P3_221$ ($D_3^6$)] that consists of helical chains of tellurium atoms in a

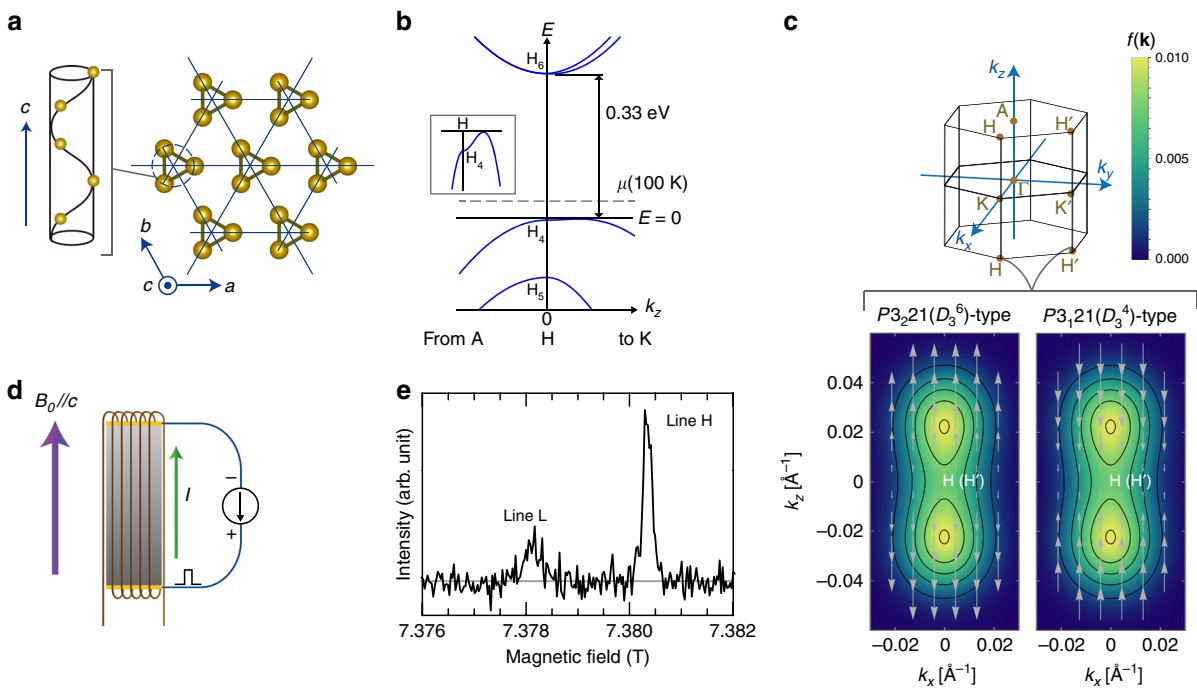

**Fig. 1** Basic properties of trigonal tellurium and experimental setup. **a** The crystal structure of the trigonal tellurium with the right-handed structure ($P3_121(D_3^4)$) consists of threefold-symmetric helical chains. **b** Schematic band structure of trigonal tellurium around the H and H' points in the first Brillion zone. The two points are related by a time reversal operation. The chemical potential at 100 K is described by the broken line $\mu(T = 100$ K). Inset: zoom on the top of the uppermost valence band. **c** The first Brillion zone and distribution of the holes at 100 K. Although the holes are not Fermi-degenerate at 100 K, they only belong to the uppermost valence band. The colours of the lower panels represent the distribution function of the holes at $T = 100$ K, $f(\mathbf{k}) = 1/[\exp\{(-E(\mathbf{k}) + \mu(100$ K$))/k_BT\} + 1]$ (where $k_B$ is the Boltzmann constant) and the lines indicate constant $f(\mathbf{k})$ contours: $f(\mathbf{k}) = 0.002, 0.004, 0.006, 0.008$ and 0.01. The arrows represent the direction and the magnitude of the spin of the electron of the uppermost valence band,

$\mathbf{s}(\mathbf{k}) = (\sim 0, \sim 0, 3Sk_z/2\sqrt{S^2k_z^2 + \Delta^2})$. The spins are almost parallel to the $c$ axis and radial-like from the H (H') point. **d** Geometry of the experiment. A coil for NMR measurements was wound around a single crystal of tellurium. A static magnetic field $B_0$ and a pulsed electric current $I$ were applied parallel to the $c$ axis. **e** $^{125}$Te-NMR spectrum of a single crystal of trigonal tellurium at 100 K under a magnetic field applied approximately parallel to the $c$ axis, in the absence of a pulsed electric current. The spectrum is plotted as a function of the effective magnetic field felt by the $^{125}$Te nuclei (see Methods)

hexagonal arrangement, as shown in Fig. 1a. Strong covalent bonds form between nearest-neighbour atoms in each chain, whereas weak van der Waals interactions act between chains. The three atoms in the unit cell are related by a threefold right-handed ($P3_121$) or left-handed ($P3_221$) screw operation along the $c$ axis. In addition, the structure has twofold rotational symmetry along the $a$, $b$, and $a + b$ axes. The trigonal structure lacks both inversion and mirror symmetry. Thus, there are two types of chiral crystals formed by either right- or left-handed helical chains.

Figure 1b illustrates the schematic band structure of $p$-type trigonal tellurium [$P3_121$ ($D_3^4$)] near the H and H' points in the Brillouin zone (Fig. 1c), around which the bottoms of the conduction bands and the tops of the valence bands are located[20, 21]. The energy dispersion of the uppermost valence band near the H and H' points is well approximated by

$$E(\mathbf{k}) = A(k_x^2 + k_y^2) + Bk_z^2 + \sqrt{S^2 k_z^2 + \Delta^2} - \Delta - E_0, \text{ where } \mathbf{k} =$$

$(k_x, k_y, k_z)$ is a wave vector measured from the H or H' points, $A = -32.6$ eV $\text{Å}^2$, $B = -36.4$ eV $\text{Å}^2$, $S = \pm 2.47$ eV Å ($-$ for $P3_121$ and $+$ for $P3_221$), $\Delta = 63$ meV, and $E_0 = 2.4$ meV[20–26]. Tellurium has spin-polarized isoenegetic surfaces as a result of its strong spin-orbit interaction and inversion asymmetry. Note that the spin texture of tellurium tends to be radial from the H (H') point, in contrast to that of a Rashba-type circular texture, i.e., the crystal symmetry of tellurium imposes the requirement that the spins on the K–H, K' –H', and H–H' lines are parallel to each line because of the threefold screw symmetry on the K–H (K' –H') lines and the twofold symmetry on the H–H' lines without any mirror symmetry. Indeed, theoretical investigations[21, 22, 27] have confirmed that the conduction bands have simple radial spin textures and that the uppermost valence band also has a radial-like, but almost $c$-axis oriented, spin texture of the electron, $\mathbf{s}(\mathbf{k}) = (\sim 0, \sim 0, 3Sk_z/2\sqrt{S^2 k_z^2 + \Delta^2})$, near the H(H') point. Note that the spin textures for right- and left-handed crystals are opposite to one another, because the two crystal structures are related by the spatial inversion. Accordingly, we expect that the applied current will induce the electronic spin polarization (anti-) parallel to the $c$ axis in $p$-type tellurium when the applied current has a $c$-axis component.

**Experimental setup**. We measured the $^{125}$Te NMR spectra of a right-handed single crystal ($P3_121$ ($D_3^4$), see the Methods section for details) at 100 K under an applied pulsed electric current. Owing to slight departure from stoichiometry, tellurium generally has a finite carrier (hole) density, which was estimated to be $5 \times 10^{15}$ cm$^{-3}$ for the present sample by Hall coefficient measurements. The present sample is in the extrinsic region at 100 K, and the density of the thermally excited carriers across the band gap is negligible in this temperature region. A static magnetic field of 7.3858 T and a pulsed current synchronized with the NMR measurement were applied parallel to the $c$ axis (screw axis), as shown in Fig. 1d (See Methods). Figure 1e shows the $^{125}$Te NMR spectrum obtained in the absence of an applied current. The spectrum consists of a low-field line (line L) around 7.37810 T and a high-field line (line H) around 7.38034 T, which are associated with one and two of the three tellurium atoms in the unit cell, respectively. If the applied magnetic field was exactly along the $c$ axis, the NMR spectrum would contain only a single line, owing to the threefold screw symmetry. The presence of two lines in Fig. 1e indicates a slight difference of $\sim 6°$ between the magnetic field direction and the $c$ axis (see Methods for details). Below, we focus on line H to discuss the current-induced electronic magnetization because of its higher signal-to-noise ratio.

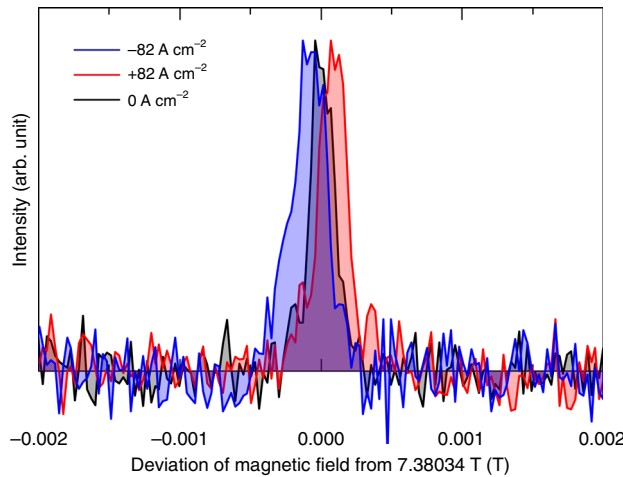

**Fig. 2** Pulse-current dependence of line H in $^{125}$Te-NMR spectrum. The NMR spectra are shown as a function of deviation of magnetic field felt by the $^{125}$Te nuclei (see Methods) from 7.38034 T. The black, red, and blue lines correspond to data obtained under pulsed electric current densities of 0, + 82, and −82 A cm$^{-2}$, respectively

**Current-induced shift of NMR spectrum**. Figure 2 shows line H for pulsed currents of 0 and $\pm 82$ A cm$^{-2}$. The shift in line position in the presence of an applied current clearly indicates the emergence of electronic magnetization. The spectral shift is approximately $10^{-1}$ mT and its sign depends on the polarity of the electric current. (The polarity of the current is defined as positive when the current and magnetic field are parallel.) Note that the spectral shape is almost unchanged by the applied current, which indicates that the current-induced electronic magnetization is spatially almost uniform. We stress that the observed current-induced shift is due to neither Joule heating nor the trivial magnetic field associated with Ampere's law (Oersted field). The former is easily ruled out by the fact that the sign of the shift depends on the polarity of the applied current. If the shift were due to Joule heating, the same shift would have been observed irrespective of the polarity. The parasitic effect of an Oersted magnetic field can also be ruled out as follows. First, for the present experimental geometry, the electric current generates a circular Oersted field about the $c$ axis; thus, the Oersted field does not alter the $c$ axis component of the local magnetic field, which determines the NMR frequency. Secondly, even though there is a slight difference between the directions of the electric current and the applied magnetic field owing to meandering of the current and/or misalignment of the sample, an Oersted field would cause only spectral broadening, and not a shift, because of the circular distribution of the field around the local current. Therefore, the observed current-induced shift cannot be explained by an Oersted field. The exclusion of these two trivial causes clearly indicates that the shift is due to a hyperfine field generated by uniform current-induced electronic magnetization.

We note that the relation between the signs of the electric current and the current-induced shift can be explained by the spin texture of the uppermost valence band of the present $P3_121$ ($D_3^4$)-type crystal (Fig. 1c). When a positive electric current is applied along the $c$ axis, the spin texture causes a negative net electronic spin polarization (i.e., a positive electronic spin magnetization), which generates a positive shift of NMR spectra because of the positive hyperfine coupling coefficient[28]. Our observations are consistent with this scenario. (We provide a rough estimate of the present current-induced magnetization in Supplementary Note 1.)

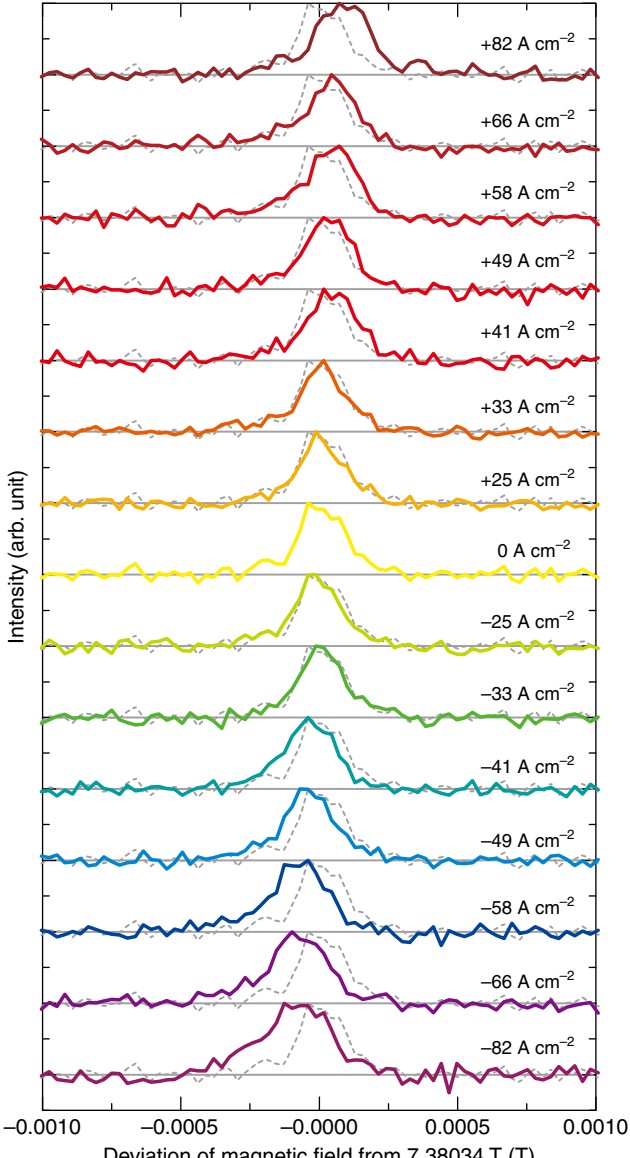

**Fig. 3** Line H in $^{125}$Te-NMR spectrum for different electric current densities. The NMR spectra are shown as a function of deviation of magnetic field felt by the $^{125}$Te nuclei (see Methods) from 7.38034 T. The values of pulsed electric current densities are $0, \pm 25, \pm 33, \pm 41, \pm 49, \pm 58, \pm 66,$ and $\pm 82$ A cm$^{-2}$. The spectrum in the absence of a pulsed electric current (dashed lines) is indicated in each row for comparison

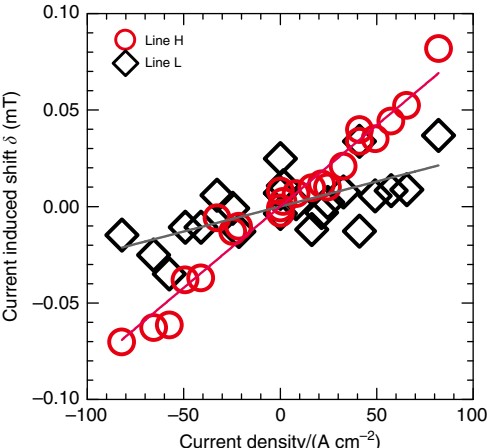

**Fig. 4** Current density dependence of current-induced shift. Red circles and black diamonds indicate the current-induced shift $\delta$ (mT) for line H and line L, respectively. The proportionality coefficient obtained by least-squares fitting is $8.4(\pm 0.4) \times 10^{-4}$ mT A$^{-1}$ cm$^2$ for line H and $2.6(\pm 0.6) \times 10^{-4}$ mT A$^{-1}$ cm$^2$ for line L

and H are estimated to be $2.6(\pm 0.6) \times 10^{-4}$ and $8.4(\pm 0.4) \times 10^{-4}$ mT A$^{-1}$ cm$^2$, respectively. In principle, the current-induced electronic magnetization for the three tellurium atoms in the unit cell should be the same. Thus, the fact that the current-induced shift is smaller for line L is very likely due to different hyperfine coupling tensor components for the tellurium nuclei associated with the two lines.

## Discussion

The observed current-induced electronic magnetization clearly represents a new class of magnetoelectric phenomena in bulk materials because the effect does not require any ferroic order. Furthermore, it is moving electrons that play a vital role in the present magnetoelectricity of tellurium, whereas it is localized electrons in the case of multiferroic materials. Moving electrons may exhibit a wide variety of magnetoelectric effects; indeed, a recent theoretical study proposed that in addition to current-induced spin magnetization, trigonal tellurium can also exhibit current-induced orbital (helical motion of holes) magnetization when an electrical current is applied parallel to the helical chain[29]. This is because an electric current along the helical axis causes the helical motion of a wave packet, which can be regarded as being the condensed matter analogue of a solenoid. Therefore, the present current-induced electronic magnetization may possess not only a spin contribution, but also a helical motion contribution. The mutual enhancement of the current-induced electronic spin and helical motion magnetization is an interesting future issue.

## Methods

**Sample preparation**. A single crystal of trigonal tellurium was obtained by repeated Bridgman growth. The sample size along the $c$ axis is approximately $L = 3.2$ mm, and the cross sectional area is $S = 0.61$ mm$^2$ ($0.71$ mm $\times 0.86$ mm). The handedness of the crystal was determined by an observation of the shapes of the etch pits produced by the slow action of the hot sulfuric acid (100 °C, 30 min) on the cleavage planes of the crystals[30]. The mobility of the holes was approximately 500 cm$^2$ V$^{-1}$ s$^{-1}$ at 100 K. A pulsed electric current was applied approximately along the $c$ axis through two electrodes placed at the top and bottom of the sample. For good electrical contact between the sample and electrodes, titanium and gold were evaporated on the sample. Copper wires were then attached using silver paste. The sample was coated with epoxy (Stycast 1266) and fixed on a glass plate so that it would not move as a result of the pulsed current. A coil with a dimension of 1.0 mm $\times$ 4.2 mm$^2$ (1.3 mm $\times$ 3.3 mm along the $c$ axis) was wound around the epoxy-coated sample for NMR measurements.

Figure 3 shows the current-density dependence of line H in the NMR spectrum. The current-induced NMR shift increases with increasing current density, but the spectral shape is almost unchanged. The slight broadening that occurs at higher current density likely arises from the nonuniform current distribution in the sample. The current density dependence of the shift is smooth and shows no threshold. This is consistent with the scenario that an imbalance between populations of oppositely spin-polarized carriers causing the current-induced electronic magnetization. We evaluated the current-induced shift of the NMR spectrum, $\delta$, under current density $I$ using centers of gravity for lines L and H, $<B(I)>_{\text{L or H}}$; namely, $\delta \equiv <B(I)>_{\text{L}} - 7.37810$ T for line L and $\delta \equiv <B(I)>_{\text{H}} - 7.38034$ T for line H. Figure 4 plots the current-induced shift $\delta$ for lines L and H against the current density. The plots are linear, and the proportionality coefficients for lines L

**NMR measurements**. The NMR spectra of $^{125}$Te (nuclear spin $I = 1/2$, gyro-magnetic ratio $\gamma = 13.454$ MHz/T) were measured at 100 K under an external magnetic field of 7.3858 T applied almost parallel to the $c$ axis. The NMR spectra were obtained as a function of the resonance frequency $f$ by Fourier transformation of the spin-echo signals following a $\pi/2–\pi$ pulse sequence. Throughout the present paper, the NMR spectra are shown as a function of the magnetic field felt by the $^{125}$Te nuclei, calculated using $B$ (T) $= f$ (MHz)$/\gamma$. The NMR pulse sequence and the pulsed electric current were synchronized such that the current pulse was switched on a sufficient length of time before the NMR $\pi/2$ pulse, and switched off well after the decay of the spin-echo signals, as shown in Supplementary Fig. 1. The polarity of the current is defined as positive when the current and the magnetic field are parallel. For an arbitrary magnetic field direction, the NMR spectrum of trigonal tellurium should have three lines with the same intensity, corresponding to the three inequivalent atoms in the unit cell. For the present sample, we verified that such a three-line spectrum appeared when the direction of the magnetic field was rotated away from the $c$ axis (Supplementary Fig. 2), which confirms that the sample is a single crystal. Accordingly, the two-line spectrum in Fig. 1e indicates that the magnetic field is applied in a direction that deviates slightly from the $c$ axis. We estimated the angular difference to be approximately 6° by comparing the two-line spectrum with the shift parameters reported in ref. [31]. When the maximum current density of 82 A cm$^{-2}$ with the pulse duration of 650 μs is applied to the sample, the temperature of the sample is estimated to have risen by 5 K at most; this estimate was obtained by calculating the product of the applied current and the voltage between the electrodes including the voltage drop due to the contact resistance. Such a temperature rise does not alter the density of the holes and the Knight shift.

**Data availability**. The data that support the plots within this paper and other findings of this study are available from the corresponding author upon reasonable request.

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

## Acknowledgements

T.I. thanks F. Kagawa and M. Hirayama for inspiring discussion at the early stage of this project. We thank T. Yasui and N. Enomoto for experimental assistance, and N. Miyakawa for his help in attaching electrodes to the sample. We also especially thank N. Miura for the sample preparation, and M. Tokunaga for the sample preparation, stimulating discussion, and helpful suggestions for improving the manuscript. This work was supported in part by JSPS KAKENHI Grant Nos. 15K13524, 25220709.

## Author contributions

T.F., Y.S., and T.I. performed the experiments and analysed the data. T.F., Y.S., and T.I. interpreted the data. K.K. prepared the single crystals for the study. T.I. planned and supervised this project. T.F. wrote the manuscript with the assistance of T.I. All authors reviewed and commented on the manuscript.

## Additional information

**Competing interests:** The authors declare no competing financial interests.

