## [Peer Review File · Nature Communications]

Reviewers' comments:

Reviewer #1 (Remarks to the Author):

Overall comment:

In the manuscript titled "Direct observation of current-induced bulk magnetization in elemental tellurium", the authors experimentally investigate current-induced magnetization of tellurium crystal using nuclear magnetic resonance (NMR) technique. The phenomenon of current-induced spin polarization (magnetization) was earlier experimentally observed in a number of bulk semiconductors by means of optical methods. Namely, current-induced spin polarization was optically detected in gyrotropic crystals such as bulk tellurium (Refs. 19, 20), wurtzite ZnSe and GaN epitaxial layers (Refs. 14, 15). Similar experiments were carried out also on strained zinc-blende GaAs and InGaAs epilayers (Ref. 13). Thus, the current-induced bulk magnetization itself is not a new issue. The new approach of the reviewed paper is application of NMR technique to study the phenomenon. But it should be noted that a theoretical consideration of this approach has been already performed by A.G. Aronov and Yu.B. Lyanda-Geller in the paper "Nuclear electric resonance and orientation of carrier spins by an electric current" [JETP Letters 50, 431–434 (1989)].

That is why the authors' claims "we show a new class of bulk magnetoelectric effects" and "this finding provides a new stage of magnetoelectricity in bulk matter" seem to be invalid. The position adopted in the work calls for a series of questions and additional remarks (see below).

Nevertheless the subject of the manuscript is very interesting. The authors carried out the experimental study of current-induced bulk magnetization by means of nuclear magnetic resonance for the first time. Phenomena that can be used to initialize, control, or detect spins in condensed matter systems are of central importance to the field of spintronics. Of particular interest in semiconductors are mechanisms that allow these tasks to be completed solely by electrical means. In this respect the manuscript is timely and after major revision could be potentially appealing to a rather broad audience.

In conclusion, I cannot recommend publication of this article in Nature Communications unless a major revision of the manuscript will be undertaken.

Additional remarks and questions:

1. Sentence "...an electric current causes uniform spin polarization owing to an imbalance between populations of up and down spins, which is called the Edelstein-Rashba effect (Ref.6)" is misleading. The same current-induced phenomenon was described earlier in Ref. 19 and in [A.G. Aronov and Yu.B. Lyanda-Geller, JETP Letters 50, 431–434 (1989)] and in a few other papers. That is why the term "Edelstein-Rashba effect" is not widely used (Google search gives only 6 mathes). The term "current-induced spin polarization (CISP)" is much more widely used for this phenomenon (Google search gives 498 mathes).

2. Sentence "We propose that an applied current can induce bulk magnetization in non-centrosymmetric materials..." is not correct in general case. There are two opportunities to transform it to the correct one: to replace "non-centrosymmetric materials" by "non-centrosymmetric materials without mirror symmetry" or by "gyrotropic materials". Here "We propose..." should be also replaced by "As it is known (Refs. 6–15, 19–20)..."

3. Sentence "...we expect that the current-induced spin polarization in tellurium would be parallel to the direction of the applied electric current." is not correct in the case of electric current directed at an angle to the c-axis of tellurium crystal. For any current direction, current-induced spin polarization in tellurium should to be parallel to the c-axis.

4. The authors should provide more explicit information concerning the experimental conditions. What type of tellurium crystal was used in the experiment (trigonal space group P3121 or P3221)? Dextrorotatory or levorotatory? Authors should discuss in the paper how the direction of the experimentally detected current-induced magnetization is related to the certain space group and the current direction. What is the position of Fermi level for the sample under investigation at operating temperature? It is desirable also to specify the hole mobility because its value indicates the crystal quality. What is the sample size along the c-axis? What is the area of the sample cross section in the perpendicular direction?

5. It is desirable to bring the Figure 1(a) into line with the certain space group. It is desirable also to bring the Figures 1(b) and 1 (c) into line with the actual position of the Fermi level (qualitatively).

6. The authors widely use term "magnetization". In the manuscript it is not always clear do they mean magnetization of free holes or magnetization of nuclei or total magnetization.

Reviewer #2 (Remarks to the Author):

In the manuscript entitled 'Direct observation of current-induced bulk magnetization in elemental tellurium', Yuri Shimokawa, et al., report on NMR measurements carried out synchronous with high-amplitude current pulses. The results are interpreted as evidence for changes to the hyperfine fields and therefore indicative of the claimed current-induced bulk magnetization. The largest of what are presumed to be hyperfine field changes amounts to approximately 0.5-0.8G in the presence of current pulses up to about 80 A/cm². Motivation for the experiments are associated with theoretical predictions related to current-induced non-equilibrium occupations of spin-split bands for fields and currents applied along the c-axis.

The experiments and results could be evidence for the stated claim, however the information provided is insufficient for an informed judgement and therefore should not be published in the current form. It could be fixable, but whether it is or not depends on details. Consider, for example, the following:

1. Missing is any specific information about sample and coil dimensions. While current densities are reported, such details would make it possible for estimates of macroscopic orbital fields. While the authors have argued against this contribution contaminating their data, it would be helpful for the readers to appreciate the scale of orbital fields.
2. The 6 degree misalignment estimate might be reasonable based on Fig. 3 in Ref. 22 (should be Phys. Stat. Sol. B, rather than Phys. Stat. Sol.), which shows the shifts for an ac (or bc) rotation. This estimate is based on the knowledge of chemical shifts in insulating tellurium. What is omitted is an estimate of the hyperfine fields without the current applied. Since a Zeeman effect would be expected, there should be something. Without that information, no estimate of the change in spin polarization is possible. And without that information, it is impossible to say that the 0.5G changes seen are reasonable.
3. Related to (2), what are the expected changes to the occupations, quantitatively, and in terms of wavevector occupations? This latter could be illustrated with an inset to Fig. 1b.
4. The carrier concentration is indicated as extrinsic. How much of what is observed is thermally excited across the 330 meV gap? If it's essentially none (compared to the 10^{15} cm^{-3}), just say so. Otherwise, the hyperfine fields are temperature-dependent. Would this have any implications for the resulting response? Related to this question, how much is the sample heating in presence of large current pulses. Although the authors are convincing that heating cannot explain the results, it would still be useful to know much the temperature is changed, and what that implies for

the carrier concentrations and thus hyperfine fields.

5. Finally, the opening paragraph motivates the interest in problems like this one. It references magnetoelectric coupling in multiferroics, and then goes on to say that this demonstrates current-induced magnetization in a system without coupled orders. It would be appropriate if somewhere the authors said something about how big the effects would have to be in order that it could be worthwhile to exploit in an application. Then, what properties would have to be tweaked to get there?

Reviewer #3 (Remarks to the Author):

The authors have performed NMR measurements on a p-type Tellurium crystal under conditions of varying current density. They claim that small current-induced shifts of a resonance line indicate charge-carrier spin polarization in the nondegenerate valence band and an effective nuclear field (making this a kind of Knight shift).

The method to use NMR to detect charge carrier spin polarization at first appears clever. While the results are weak (max shift 1G), they are suggestive of a scenario consistent with their claims. However, I find many inadequacies in this paper.

Although NMR can be done at low field strengths, here the authors use a large field, $>7T$. This opens the possibility that a simple band picture does not apply (formally, because the vector potential breaks translational symmetry). Are Shubnikov-deHaas oscillations present under these conditions?

In light of the expected large g-factor in the uppermost valence band (from SOC-split orbitals), a large B-field makes one consider the reciprocal effect where a magnetic field induces an electric field...

Can the authors use a lower magnetic field (I realize this requires a lower RF frequency), and is the relationship between line shift and current density the same?

What is the temperature dependence? What happens if the current is not along the c axis? Can the experiment be repeated for trigonal Se?

Fig 1c shows separate Fermi surfaces near the $H(H')$ points. However, Doi (Ref 17 and its companion paper in 1970 in the same journal) gives a depth of the uppermost valence band of 1.1meV. At 100K, this is feature is irrelevant. Also, the description of this panel on p. 4 is incorrect -- the conduction band has a radial spin texture, but the valence band does not. Please read Ref. 18 carefully, and/or diagonalize the Hamiltonian given by Doi and examine the expectation values of spin operators for the highest-energy eigenstate.

No comparison to numerical estimates from theory is presented here. In other words, does 1G shift in $\sim 100 \text{ A/cm}^2$ make sense? This will require some knowledge of the Knight shift but that can presumably come from temperature dependence of carrier spin polarization in the absence of a current. It also requires knowledge of Fermi surface shift but that is easily obtainable in linear response from the conductivity and effective mass (to determine scattering time). Follow a procedure similar to that given in Phys. Rev. B 93, 220404(R) (2016), using the appropriate dispersion expansion.

Text from line 137-140 is nonsensical to me.

Replies to Reviewer #1's comments

We thank Reviewer 1 for reviewing our manuscript and providing useful suggestions on how we could improve it. In particular, his/her expertise in current-induced spin polarization has helped us make useful revisions to the manuscript. We provide replies to each point in turn below.

[Comment 1-0]

In the manuscript titled “Direct observation of current-induced bulk magnetization in elemental tellurium”, the authors experimentally investigate current-induced magnetization of tellurium crystal using nuclear magnetic resonance (NMR) technique. The phenomenon of current-induced spin polarization (magnetization) was earlier experimentally observed in a number of bulk semiconductors by means of optical methods.

Namely, current-induced spin polarization was optically detected in gyrotropic crystals such as bulk tellurium (Refs. 19, 20), wurtzite ZnSe and GaN epitaxial layers (Refs. 14, 15). Similar experiments were carried out also on strained zinc-blende GaAs and InGaAs epilayers (Ref. 13). Thus, the current-induced bulk magnetization itself is not a new issue. The new approach of the reviewed paper is application of NMR technique to study the phenomenon. But it should be noted that a theoretical consideration of this approach has been already performed by A.G. Aronov and Yu.B. Lyanda-Geller in the paper “Nuclear electric resonance and orientation of carrier spins by an electric current” [JETP Letters 50, 431–434 (1989)].

That is why the authors' claims “we show a new class of bulk magnetoelectric effects” and “this finding provides a new stage of magnetoelectricity in bulk matter” seem to be invalid. The position adopted in the work calls for a series of questions and additional remarks (see below).

Nevertheless the subject of the manuscript is very interesting. The authors carried out the experimental study of current-induced bulk magnetization by means of nuclear magnetic resonance for the first time. Phenomena that can be used to initialize, control, or detect spins in condensed matter systems are of central importance to the field of spintronics. Of particular interest in semiconductors are mechanisms that allow these tasks to be completed solely by electrical means. In this respect the manuscript is timely and after major revision could be potentially appealing to a rather broad audience.

In conclusion, I cannot recommend publication of this article in Nature Communications unless a major revision of the manuscript will be undertaken.

[Reply 1-0]

This comment helped us realize that our original manuscript lacked sufficient explanation.

First of all, the term “bulk” is used in different ways in different scientific fields, and the meaning of the term “bulk” that we intended was different from that used by the reviewer. In our manuscript, the term “bulk” refers to an infinite system without boundary effects; thus “bulk materials” and “bulk properties” mean materials and properties, respectively, that are free of surface effects including the local strain caused by a heterointerface. Accordingly, we intended not to include thin films in the term “bulk.” Although ZnSe, GaN and GaAs–InGaAs epitaxial layers have been found to exhibit current-induced magnetization, they are thin films on substrates, which are not bulk materials in our definition. In this context, the current-induced magnetization in genuine bulk materials has not yet been established. The only pioneering work is an optical activity measurement in bulk elemental tellurium, which we have appreciated and described in our manuscript. In this particular study, a current-induced spin polarization was discussed through the current-induced modulation of the optical activity of tellurium. In contrast, the present NMR measurement directly observed the current-induced magnetization. Our work is, to the best of our knowledge, the first *direct* observation of current-induced magnetization in a genuine bulk material.

We are sorry that the original manuscript did not adequately explain our standpoint this definition of the term “bulk”. Thus, the original manuscript was misleading, because “bulk” is used in a different meaning in a part of the spintronics field. In order to clarify this point, we have added the definition of the term “bulk” into the manuscript and explained that it does not include thin films. We really appreciate the reviewer’s comment. We are happy to resolve the misleading point in this revision.

In lines 42 and 43 of the original manuscript, the text

“This effect has been experimentally observed for surfaces⁹, interfaces¹⁰⁻¹⁴, and epilayers¹⁵⁻¹⁷”

has been replaced by

“This effect has been experimentally observed for surfaces⁹, interfaces¹⁰⁻¹⁴, and epilayers¹⁵⁻¹⁷, which are not bulk systems.”

and the text

“(In this study, the term “bulk” refers to an infinite system free of boundary effects including the local strain caused by a heterointerface)”

has been added into page 3 of the revised manuscript.

The reviewer also pointed out that the strategy of using NMR to study current-induced magnetization has been already suggested by A.G. Aronov *et al.* (*JETP Letters* **50**, 431–434 (1989)). Their paper theoretically discussed the possibility that the application of an *alternating* electric field to a system with spin-split energy bands causes a nuclear magnetic transition *depending on the spin relaxation of the carriers*. Although their study describes an as-yet unobserved current-induced phenomena, their argument is a little different from that presented in our work. We used NMR as an experimental probe that can make direct, sensitive, and microscopic detections of local static electronic magnetizations. In addition, the current-induced electronic spin magnetization discussed in our manuscript is induced by a *static* electric field (*i.e.*, a *static* electric current) and does not require the spin relaxation of carriers. However, we agree with the reviewer that the pioneering work by Aronov *et al.* should be referred to in the manuscript, and we have revised as follows.

In lines 41 and 42 of the original manuscript, the text

“...which is called Edelstein-Rashba effect⁶”

has been replaced by the text

“...which is called current-induced spin polarization⁶⁻⁸”

with an additional reference for

7. Aronov, A. G. & Lyanda-Geller, Y. B. Nuclear electric resonance and orientation of carrier spins by an electric field. *JETP Lett.* **50**, 431–434 (1989).

[Comment 1-1]

Sentence "...an electric current causes uniform spin polarization owing to an imbalance between populations of up and down spins, which is called the Edelstein-Rashba effect (Ref.6)" is misleading. The same current-induced phenomenon was described earlier in Ref. 19 and in [A.G. Aronov and Yu.B. Lyanda-Geller, JETP Letters 50, 431–434 (1989)] and in a few other papers. That is why the term "Edelstein-Rashba effect" is not widely used (Google search gives only 6 mathes). The term "current-induced spin polarization (CISP)" is much more widely used for this phenomenon (Google search gives 498 mathes).

[Reply 1-1]

Following the reviewer's suggestion in the comment, we have revised the manuscript, as described below.

In lines 41 and 42 of the original manuscript, the text

"...which is called Edelstein-Rashba effect⁶"

has been replaced by the text

"...which is called current-induced spin polarization⁶⁻⁸"

with an additional reference for

7. Aronov, A. G. & Lyanda-Geller, Y. B. Nuclear electric resonance and orientation of carrier spins by an electric field. *JETP Lett.* **50**, 431–434 (1989).

[Comment 1-2]

Sentence "We propose that an applied current can induce bulk magnetization in non-centrosymmetric materials..." is not correct in general case. There are two opportunities to transform it to the correct one: to replace "non-centrosymmetric materials" by "non-centrosymmetric materials without mirror symmetry" or by "gyrotropic materials". Here "We propose..." should be also replaced by "As it is known (Refs. 6–15, 19–20)..."

[Reply 1-2]

The former formulation made by the reviewer in the comment here makes it sound as if an applied current can induce bulk magnetization in non-centrosymmetric materials *without* mirror symmetry, but cannot induce it generally in non-centrosymmetric materials *with* mirror symmetry; however, this is incorrect. For example, let us consider bulk BiTeI ($P3m1$), which is a non-centrosymmetric material *with* mirror symmetry that has giant Rashba-type spin splitting bands [Ishizaka *et al.* *Nature Materials* **10**, 521 (2011)]. In this system, an applied electric current parallel to the x axis can induce a spin polarization parallel to the y axis due to its circular spin texture. Thus, we do not believe that the text should be changed in the way proposed.

With respect to the latter of the two suggestions, our original description does not disregard previous pioneering studies; rather, to the best of our knowledge, all previous works about current-induced spin polarizations have been limited to non-bulk systems (*i.e.*, surfaces, interfaces, and epilayers) aside from the optical studies of bulk tellurium [8, 26] (again, we note that the meaning of the term "bulk" that we use is different from that used by the reviewer, and we have revised the manuscript accordingly; please see Reply 1-0). These optical studies on

tellurium are innovative, and thus we have described them in detail in our manuscript. In those studies, current-induced spin polarization was discussed through the current-induced modulation of the optical activity of tellurium; by contrast, the present NMR measurements directly observed the current-induced magnetization for the first time. In order to explain the position of our work and pay due respect to the pioneering optical works, we have revised the text in our manuscript, as described below.

In line 44 of the original manuscript, the text

“We propose that ...”

has been replaced by

“We demonstrated that ...”.

[Comment 1-3]

Sentence “...we expect that the current-induced spin polarization in tellurium would be parallel to the direction of the applied electric current.” is not correct in the case of electric current directed at an angle to the c -axis of tellurium crystal. For any current direction, current-induced spin polarization in tellurium should to be parallel to the c -axis.

[Reply 1-3]

We know that the current-induced spin polarization in tellurium should, in principle, be parallel to the c axis when the lowest order effect of the spin splitting is considered.

We intended to state that the current-induced spin polarization in tellurium would be *almost* parallel to the direction of the applied electric current *in the present experimental setting ($I \parallel c$)*. However, this comment led us to notice that the original sentence was misleading and unclear; we have revised the manuscript as follows.

In lines 71 and 72 of the original manuscript, the text

“Accordingly, we expect that the current-induced spin polarization in tellurium would be parallel to the direction of the applied electric current.”

has been replaced by

“Accordingly, we expect that the applied current will induce the electronic spin polarization (anti-)parallel to the c axis in p -type tellurium, when the applied current has a c -axis component.”

[Comment 1-4]

The authors should provide more explicit information concerning the experimental conditions.

(a) What type of tellurium crystal was used in the experiment (trigonal space group P3121 or P3221)? Dextrorotatory or levorotatory? Authors should discuss in the paper how the direction of the experimentally detected current-induced magnetization is related to the certain space group and the current direction.

(b) What is the position of Fermi level for the sample under investigation at operating temperature?

(c) It is desirable also to specify the hole mobility because its value indicates the crystal quality. What is the sample size along the c -axis? What is the area of the sample cross section in the perpendicular direction?

[Reply1-4]

(a) The crystals used in the present study had a $P3_121 (D_3^4)$ structure, which we determined by observing the etch pits; the forms of the etch pits depend on the crystal handedness [A. Koma *et al. phys. stat. sol.* **40**, 239 (1970)]. It has been reported that $P3_121(D_3^4)$ -type crystals are levorotatory [Blakemore *et al. Journal of Applied Physics* **32**, 745 (1961), A. Koma *et al. phys. stat. sol.* **40**, 239 (1970), Brown *et al. Acta. Cryst.* **A52**, 408-412 (1996), Y. Tanaka *et al. J. Phys.: Condens. Matter* **22** 122201 (2010)].

The relation between the signs of the electric current and the current-induced shift can be explained by the spin texture of the uppermost valence band of the present $P3_121(D_3^4)$ -type crystal (Fig. 1c). When a positive electric current is applied in line with the c axis, the spin texture causes a negative net electronic spin polarization (the positive electronic spin magnetization), which generates a positive shift of NMR spectra because of the positive hyperfine coupling coefficient (Selbach *et al. Phys. Rev. B* **19**, 4435–4443 (1979)). Our observations are consistent with this scenario.

(b) If we consider that an acceptor level is located 1.3 meV above the top of the uppermost valence band [Couder *et al, Phys. Rev. B* **7** 4373, (1973)] and that the concentration of the acceptor is approximately $5 \times 10^{15} \text{ cm}^{-3}$, then a Fermi level of 39.5 meV was produced at 100 K based on the band dispersion $E(\mathbf{k}) = A(k_x^2 + k_y^2) + Bk_z^2 + \sqrt{S^2 k_z^2 + \Delta^2} - \Delta - E_0$, where $\mathbf{k} = (k_x, k_y, k_z)$ is a wave vector from the H or H' points, $A = -32.6 \text{ eV \AA}^2$, $B = -36.4 \text{ eV \AA}^2$, $|S| = 2.47 \text{ eV \AA}$, $\Delta = 63 \text{ meV}$, and $E_0 = 2.4 \text{ meV}$ [Betbeder-Matibet *et al. Phys. Stat. Sol.* **36**, 573–586 (1969). Doi *et al. J. Phys. Soc. Japan* **28**, 36–43 (1970). Braun *et al. Phys. Status Solid (b)* **53**, 635–650 (1972). Stolze *et al. Phys. Status Solidi (b)* **82**, 457–466 (1977).].

(c) The hole mobility was approximately $500 \text{ cm}^2 \text{ V}^{-1} \text{ s}^{-1}$, which was estimated from Hall measurements. The sample size along the c axis was approximately $L = 3.2 \text{ mm}$, and a cross-sectional area, $S = 0.61 \text{ mm}^2$ ($0.71 \text{ mm} \times 0.86 \text{ mm}$), was used.

We have added all of the above analyses and discussions in the revised manuscript.

The text

“We measured ^{125}Te NMR spectra of a single crystal at 100 K under an applied pulsed electric current.”

in lines 79 and 80 in the original manuscript has been replaced by

“We measured the ^{125}Te NMR spectra of a right-handed single crystal ($P3_121 (D_3^4)$), see the Methods section for details) at 100 K under an applied pulsed electric current.”

The text

“Owing to slight departure from stoichiometry, tellurium generally has a finite carrier (hole) density, which was estimated to be $5 \times 10^{15} \text{ cm}^{-3}$ for the present sample by Hall coefficient measurements.”

in lines 58–60 of the original manuscript has been moved to page 6 in the revised manuscript.

The text

“We note that the relation between the signs of the electric current and the current-induced shift can be explained by the spin texture of the uppermost valence band of the present $P3_121(D_3^4)$ -type crystal (Fig. 1c). When a positive electric current is applied along the c axis, the spin texture causes a negative net electronic spin polarization (i.e., a positive electronic spin magnetization), which generates a positive shift of NMR spectra because of the positive hyperfine coupling coefficient²⁴. Our observations are consistent with this scenario.”

has been added to page 8 of the revised manuscript, and the following reference has also been added

27. Selbach, H., Kanert, O. & Wolf, D. NMR investigation of the diffusion and conduction properties of the semiconductor tellurium. I. Electronic properties. *Phys. Rev. B* **19**, 4435–4443 (1979).

The text

“The sample size along the c axis is approximately $L = 3.2$ mm, and the cross-sectional area is $S = 0.61$ mm² (0.71 mm \times 0.86 mm). The handedness of the crystal was determined by an observation of the shapes of the etch pits produced by the slow action of the hot sulfuric acid (100 °C, 30 min) on the cleavage planes of the crystals²⁹. The mobility of the holes was approximately 500 cm² V⁻¹ s⁻¹ at 100 K.”

has been added to the Methods section, along with the reference for

29. Koma, A., Takimoto, E. & Tanaka, S. Etch Pits and Crystal Structure of Tellurium. *Phys. Stat. Sol.* **40**, 239–248 (1970).

Lastly, Figs. 1b and 1c, and their legends, have been revised so as to indicate the correct distribution of the holes at 100 K.

[Comment 1-5]

It is desirable to bring the Figure 1(a) into line with the certain space group. It is desirable also to bring the Figures 1(b) and 1 (c) into line with the actual position of the Fermi level (qualitatively).

[Reply 1-5]

Following the suggestion made in this comment, we have revised the manuscript. Specifically, we have revised Fig. 1. The revised figure contains the crystal structure and space group of the present crystal, and the actual position of the Fermi level (i.e., the chemical potential).

In addition, in the legend of Figure 1, the text

“Crystal structure of trigonal tellurium consists of threefold-symmetric helical chains.”

has been replaced by

“The crystal structure of the trigonal tellurium with the right-handed structure ($P3_121(D_3^4)$) consists of threefold-symmetric helical chains.”

The text

“c, First Brillouin zone and spin-polarized Fermi surfaces of p-type tellurium with a low hole density ($n_H < 10^{17} \text{ cm}^{-3}$). The spins on the hole pockets are almost parallel to the c axis, in radial fashion from the H (H') point.”

has been replaced by

“c, The first Brillouin zone and distribution of the holes at 100 K. Although the holes are not Fermi-degenerate at 100 K, they only belong to the uppermost valence band. The colours of the lower panels represent the distribution function of the holes at $T = 100 \text{ K}$, $f(\mathbf{k}) = 1/[\exp\{(-E(\mathbf{k}) + \mu(100 \text{ K}))/k_B T\} + 1]$ (where k_B is the Boltzmann constant), and the lines indicate constant $f(\mathbf{k})$ contours: $f(\mathbf{k}) = 0.002, 0.004, 0.006, 0.008$ and 0.01 . The arrows represent the direction and the magnitude of the spin of the electron of the uppermost valence band, $\mathbf{s}(\mathbf{k}) = (\sim 0, \sim 0, 3Sk_z / 2\sqrt{S^2 k_z^2 + \Delta^2})$. The spins are almost parallel to the c axis and radial-like from the H (H') point.”

And the text

“The chemical potential at 100 K is described by the broken line $\mu(T = 100 \text{ K})$.”

has been added.

[Comment 1-6]

The authors widely use term “magnetization”. In the manuscript it is not always clear do they mean magnetization of free holes or magnetization of nuclei or total magnetization.

[Reply 1-6]

Following the suggestion made in this comment, we have replaced the term “magnetization” with “electronic magnetization” when we feel it is misleading to use just “magnetization” in our manuscript.

Replies to Reviewer #2's comments

We thank Reviewer 2 for taking the time to review our manuscript and provide comments on it. His/her constructive suggestions have helped us to improve the quality of our manuscript. Point-by-point replies are provided below.

[Comment 2-1]

1. Missing is any specific information about sample and coil dimensions. While current densities are reported, such details would make it possible for estimates of macroscopic orbital fields. While the authors have argued against this contribution contaminating their data, it would be helpful for the readers to appreciate the scale of orbital fields.

[Reply2-1]

Following the reviewer's suggestion in the comment, we have revised the manuscript, as described below.

The text

“The sample size along the c axis is approximately $L = 3.2$ mm, and the cross-sectional area is $S = 0.61$ mm² (0.71 mm \times 0.86 mm). The handedness of the crystal was determined by an observation of the shapes of the etch pits produced by the slow action of the hot sulfuric acid (100 °C, 30 min) on the cleavage planes of the crystals²⁹. The mobility of the holes was approximately 500 cm² V⁻¹ s⁻¹ at 100 K.”

has been added to the Methods section, along with the reference for

29. Koma, A., Takimoto, E. & Tanaka, S. Etch Pits and Crystal Structure of Tellurium. *Phys. Stat. Sol.* **40**, 239–248 (1970).

The size of a coil (1.0 mm \times 4.2 mm² (1.3 mm \times 3.3 mm along the *c* axis)) has also been added in the Methods section.

[Comment 2-2 & 2-3]

2. The 6 degree misalignment estimate might be reasonable based on Fig. 3 in Ref. 22 (should be Phys. Stat. Sol. B, rather than Phys. Stat. Sol.), which shows the shifts for an ac (or bc) rotation. This estimate is based on the knowledge of chemical shifts in insulating tellurium. What is omitted is an estimate of the hyperfine fields without the current applied. Since a Zeeman effect would be expected, there should be something. Without that information, no estimate of the change in spin polarization is possible. And without that information, it is impossible to say that the 0.5G changes seen are reasonable.

3. Related to (2), what are the expected changes to the occupations, quantitatively, and in terms of wavevector occupations? This latter could be illustrated with an inset to Fig. 1b.

[Reply 2-2 & 2-3]

(This Reply is almost the same to Reply 3-5 because the Comment 2-2 & 2-3 and Comment 3-5 are essentially the same.)

As the reviewer commented, we appreciate the importance of (1) estimation of the amount of the current-induced magnetization from the current-induced NMR shift observed and the hyperfine coupling and (2) comparison of the estimated magnetization with the result of the theoretical calculation that the reviewer #3 suggested. However, at present, it is impossible to estimate the amount of the induced magnetization because there is no reliable source of information on the hyperfine coupling in elemental trigonal tellurium. In other words, reliable K (Knight shift)– χ (spin susceptibility) analyses have not been achieved, despite the long length of time of the elemental tellurium study. This is because elemental trigonal tellurium is a semiconductor with a band gap E_g of ~ 330 meV, which is much larger than room temperature, and thus the spin susceptibility is too small to be detected. Indeed, the total magnetic susceptibility does not show a temperature dependence in non-doped pure crystals below room temperature (Fukuroi *et al. J. Jap. Inst. Metals* **19**, 118–122 (1955) and Fisher *et al. J. Phys. Chem. Solids* **17**, 246–253 (1961)); the experimental value of the spin susceptibility is, therefore, not available.

Nevertheless, we will provide a rough and tentative estimate of the current-induced magnetization. In Ref. 27 (Selbach, H., Kanert, O. & Wolf, D. *Phys. Rev. B* **19**, 4435–4443 (1979)), the Knight shift data of the intrinsic region (350–700 K) is reported. We estimated the hyperfine coupling to be 5.4×10^3 T/ μ_B (where μ_B is the Bohr magneton) by using the reported Knight shift data and theoretically calculated spin susceptibility; the details are provided below.

We calculate the high-temperature spin susceptibility under a magnetic field along the c axis, according to the procedure discussed in Ref. 27. First, we adopt the approximate form of the spin susceptibility of trigonal tellurium caused by non-degenerate electrons and holes thermally excited across the band gap: $\chi_s = [\mu_B^2 n_e(T) + (2\mu_B)^2 n_h(T)] / k_B T$, where $n_e(T)$ and $n_h(T)$ are the density of the electrons and holes, respectively. Next, we assume the density of the electrons and holes to be $n_e(T) = n_h(T) = (2/h^3)(2\pi k_B T)^{3/2} (m_e^* m_h^*)^{3/4} e^{-E_g/2k_B T}$, where h is the Planck constant, and m_e^* and m_h^* are the density-of-states effective mass of the conduction and valence bands, respectively. By substituting $m_e^* = 0.091m_0$ and $m_h^* = 0.143m_0$ (where m_0 is the free electron mass) (Shinno *et al. J. Phys. Soc. Japan* **35**, 525–533, (1973)), we obtain $\chi_s = \mu_B \times [7.50 \times 10^{14} \text{ (cm}^{-3} \text{ T}^{-1} \text{ K}^{-1/2})] \times [T(\text{K})]^{1/2} \times \exp(-E_g / 2k_B T)$. Lastly, we compare this susceptibility with the reported Knight shift, $K = [1.38 \times 10^{-4} \text{ (K}^{-1/2})] \times [T(\text{K})]^{1/2} \times \exp(-E_g / 2k_B T)$; this allows us to obtain a hyperfine coupling of 5.4×10^3 T/ μ_B . We note that this value does not represent the hyperfine coupling of the uppermost valence band, but rather it is the average value of those of the uppermost valence band and the conduction bands, which can be different. In addition, we neglect the effect of the temperature dependence of the orbital magnetism, which can also contribute to the temperature dependence of the Knight shift. As a result, the estimated hyperfine coupling is not very reliable.

Nevertheless, if we adopt this hyperfine coupling value (*i.e.*, 5.4×10^3 T/ μ_B), the spectral shift of ~ 0.7 Gauss (under 82 A cm^{-2}) observed in the present study yields a magnetization of $1.3 \times 10^{-8} \mu_B$ per site.

Next, we try to compare the above rough estimation of the current-induced magnetization with a theoretical calculation. When an electric field is applied along the c (z) axis, a current-induced spin polarization $\langle s_z \rangle$ and an electric current density $\langle j_z \rangle$ can be expressed by a Boltzmann transport equation approach in the relaxation-time approximation (Li *et al. Phys. Rev. B* **93**, 220404 (2016)):

$$\langle s_z \rangle = \frac{3}{2} V \int_{BZ} \frac{d\mathbf{k}}{(2\pi)^3} s_z^h(\mathbf{k}) v_z^h(\mathbf{k}) e E_z \tau \left(\frac{\partial f_0^h}{\partial \varepsilon} \right)_{\varepsilon=E^h(\mathbf{k})},$$

$$\langle j_z \rangle = -e \int_{BZ} \frac{d\mathbf{k}}{(2\pi)^3} (v_z^h(\mathbf{k}))^2 e E \tau \left(\frac{\partial f_0^h}{\partial \varepsilon} \right)_{\varepsilon=E^h(\mathbf{k})},$$

where V is a system volume, e (>0) is the elementary charge, \mathbf{k} is the wave vector of a hole, $s_z^h(\mathbf{k})$ is the z -axis component of the spin of a hole at \mathbf{k} , $v_z^h(\mathbf{k})$ is the z -axis component of the group velocity of a hole at \mathbf{k} , E_z is the z -axis component of an electric field, τ is the scattering time, $E^h(\mathbf{k})$ is the energy dispersion of a hole, $(\partial f_0^h / \partial \varepsilon)_{\varepsilon=E^h(\mathbf{k})}$ is a derivative of the equilibrium distribution function of the holes at $E^h(\mathbf{k})$ with respect to an energy, and BZ denotes the first Brillouin zone. If the energy dependence of τ is neglected, then the following equation is obtained:

$$\langle s_z \rangle = -\frac{3}{2} V \frac{\int_{BZ} d\mathbf{k} s_z^h(\mathbf{k}) v_z^h(\mathbf{k}) \left(\frac{\partial f_0^h}{\partial \varepsilon} \right)_{\varepsilon=E^h(\mathbf{k})}}{e \int_{BZ} d\mathbf{k} (v_z^h(\mathbf{k}))^2 \left(\frac{\partial f_0^h}{\partial \varepsilon} \right)_{\varepsilon=E^h(\mathbf{k})}} \langle j_z \rangle.$$

Note that the value of $eE\tau$ is absent in this equation. Below, we use the actual chemical potential at 100 K, $\langle j_z \rangle = 82 \text{ A cm}^{-2}$, $E^h(\mathbf{k}) = -E(-\mathbf{k}) = -[A(k_x^2 + k_y^2) + Bk_z^2 + \sqrt{S^2 k_z^2 + \Delta^2} - \Delta - E_0]$, $s_z^h(\mathbf{k}) = -s_z(-\mathbf{k}) = (3/2) \times S k_z / \sqrt{S^2 k_z^2 + \Delta^2}$ and $v_z^h(\mathbf{k}) = -(2Bk_z + S^2 k_z / \sqrt{S^2 k_z^2 + \Delta^2}) / \hbar$, where $\mathbf{k} = (k_x, k_y, k_z)$ is a wave vector from the H or H' points, $A = -32.6 \text{ eV \AA}^2$, $B = -36.4 \text{ eV \AA}^2$, $S = -2.47 \text{ eV \AA}$ (for $P3_121$), $\Delta = 63 \text{ meV}$, and $E_0 = 2.4 \text{ meV}$. As a result, the density of the current-induced spin polarization is calculated to be $\langle s_z \rangle / V = -4.1 \times 10^{13} \text{ cm}^{-3}$. The current-induced magnetization per tellurium atom M_{atom} is calculated to be $M_{\text{atom}} = g_{J=3/2} \mu_B \langle s_z \rangle \times V_{\text{atom}}^{-1} \sim 1.9 \times 10^{-9} \mu_B$, where $g_{J=3/2} = -4/3$ is the Landé g -factor of $J=3/2$ ($S=1/2$, $L=1$), and $V_{\text{atom}} = 34 \text{ \AA}^3$ is the atomic volume of tellurium. This theoretically calculated result, $\sim 10^{-9} \mu_B$ per site, is comparable to the above estimation obtained from the NMR shift, even though both the estimations are rough.

In spite of the rough agreement between the experimental estimation and the theoretical calculation, we emphasize again that the hyperfine coupling value described above is not very reliable and may contain considerable uncertainties. Thus, we do not believe that the above unreliable estimation should be included in the main manuscript. Instead, we show the above discussion in the Supplementary Information in the revised manuscript. We believe that the most important point of the present study is a qualitative one, i.e., direct proof of current-induced magnetization in a genuine bulk material is provided. The further quantitative estimation is beyond the scope of the present study.

Because of the non-degenerate distribution of the holes at 100 K, the current-induced changes to the occupations cannot be simply represented by the wave-number offset of the Fermi surfaces.

A typo in the Reference section of the original manuscript has been fixed in the revised manuscript as a result of these comments.

We have also revised the manuscript as follows.

The text

“(We provide a rough estimate of the present current-induced magnetization in Supplementary Information.)”

has been added in page 8 in the revised manuscript.

We have also added a new section into the Supplementary Materials in order to explain all of the above discussions, and we have added the following references:

- S1. Fukuroi, T. & Yasuhara, K. On the Magnetic Susceptibility of Tellurium and the Magnetic Anisotropy of its Single Crystal. *J. Jap. Inst. Metals* **19**, 118–122 (1955).
- S2. Fisher, G. & Hedgcock, F. T. Magnetic susceptibility and galvanomagnetic effects in pure and P-type tellurium. *J. Phys. Chern. Solids* **17**, 246–253 (1961).
- S3. Shinno, H., Yoshizaki, R., Tanaka, S., Doi, T. & Kamimura, H. Conduction Band Structure of Tellurium. *J. Phys. Soc. Japan* **35**, 525–533 (1973).
- S4. Li, P. & Appelbaum, I. Interpreting current-induced spin polarization in topological insulator surface states. *Phys. Rev. B* **93**, 220404 (2016).

[Comment 2-4]

(a) The carrier concentration is indicated as extrinsic. How much of what is observed is thermally excited across the 330 meV gap? If it's essentially none (compared to the 10^{15} cm^{-3}), just say so. Otherwise, the hyperfine fields are temperature-dependent. Would this have any implications for the resulting response? Related to this question, how much is the sample heating in presence of large current pulses. Although the authors are convincing that heating cannot explain the results, it would still be useful to know much the temperature is changed, and what that implies for the carrier concentrations and thus hyperfine fields.

[Reply 2-4]

The present sample is in the extrinsic region at 100 K; indeed, the density of the holes thermally excited across the 330-meV band gap is estimated to be less than 10^{11} cm^{-3} at 100 K. In addition, we estimated the heating of the sample by calculating the product of the applied current and the voltage between the electrodes including the voltage drop due to the contact resistance. The temperature rise of the sample was, at most, 5 K when the maximum current density of 82 A cm^{-2} with the pulse duration of $650 \mu\text{s}$ was applied to the sample. The increase in the temperature from 100 K to 105 K only excited a negligible number of holes across the 330-meV band gap; thus, the Knight shift was not affected by the heating. We have revised the manuscript as follows.

In page 6 of the revised manuscript, the text

“The present sample is in the extrinsic region at 100 K, and the density of the thermally excited carriers across the band gap is negligible in this temperature region.”

has been added.

Furthermore, in the Methods section in the revised manuscript, the text

“When the maximum current density of 82 A cm^{-2} with the pulse duration of $650 \mu\text{s}$ is applied to the sample, the temperature of the sample is estimated to have risen by 5 K at most; this estimate was obtained by calculating the product of the applied current and the voltage between the electrodes including the voltage drop due to the contact resistance. Such a temperature rise does not alter the density of the holes and the Knight shift.”

has been added.

[Comment 2-5]

Finally, the opening paragraph motivates the interest in problems like this one. It references magnetoelectric coupling in multiferroics, and then goes on to say that this demonstrates current-induced magnetization in a system without coupled orders. It would be appropriate if somewhere the authors said something about how big the effects would have to be in order that it could be worthwhile to exploit in an application. Then, what properties would have to be tweaked to get there?

[Reply 2-5]

The reviewer seems to suggest that an issue is how large the present magnetoelectric effect is. However, the important point is that the current-induced bulk magnetization in tellurium is completely different from the magnetoelectric effect in multiferroics, in terms of their origins. This new type of magnetoelectric effect in tellurium itself is of worth as a new emergent phenomenon in bulk condensed matter. Furthermore, the research of the present current-induced bulk magnetization has only just begun; thus, we are afraid that it is too early to discuss potential applications, at the moment, though we really think that the reviewer's viewpoint will be important in the future.

Replies to Reviewer #3's comments

We would like to thank Reviewer 3 for his/her careful reading of our manuscript and his/her useful suggestions. His/her deep insight into spin-split energy band systems helped us improve our manuscript. Point-by-point replies are provided below.

[Comment 3-1]

Although NMR can be done at low field strengths, here the authors use a large field, $>7T$. This opens the possibility that a simple band picture does not apply (formally, because the vector potential breaks translational symmetry). Are Shubnikov-deHaas oscillations present under these conditions?

[Reply 3-1]

Although Shubnikov-de Haas (SdH) oscillations have been observed below several K under magnetic field above ~ 7 T, there is no report of SdH oscillations at 100 K. Indeed, the Dingle temperature (T_D) is expected to be less than 1 K for the present crystal which has $n \sim 5 \times 10^{15} \text{ cm}^{-3}$ and $\mu \sim 500 \text{ cm}^2 \text{ V}^{-1} \text{ s}^{-1}$ at 100 K (Braun *et al. phys. stat. sol. (b)* **53**, 635 (1972)). Thus we think that we do not have to pay attention to this effect at 100 K, and a simple band picture is sufficient to discuss in the present experimental conditions (of course, experiments under lower magnetic fields would have been more appropriate, if they were possible. However, as explained in Reply 3-2, they are technically quite difficult.)

[Comment 3-2]

In light of the expected large g-factor in the uppermost valence band (from SOC-split orbitals), a large B-field makes one consider the reciprocal effect where a magnetic field induces an electric field...

Can the authors use a lower magnetic field (I realize this requires a lower RF frequency), and is the relationship between line shift and current density the same?

[Reply3-2]

NMR measurements at lower magnetic fields are difficult to perform, due to technical reasons. The ^{125}Te -NMR signal used in the present study was weak, because of both its small gyromagnetic ratio ($\gamma = 13.454 \text{ MHz T}^{-1}$) and the low natural abundance of this isotope (7%). The more important point is that the present measurements were performed under a pulsed strong electric current, which created a significant amount of noise. We confess that it took over three months from when the experimental setup was ready for the obtaining of the present experimental data. In general, the time required to keep sufficient signal-to-noise ratio is proportional to B_0^4 (where B_0 is an external magnetic field). If we were to try to perform the present experiments under a magnetic field whose strength was half of that used in the present study, it would take us four years. This is the reason why we performed the present experiments under ~ 7 T. Although it is important to conduct the present experiment under a lower magnetic field, it would require technical innovations to be made; such innovations are beyond the scope of the present study.

[Comment 3-3]

What is the temperature dependence? What happens if the current is not along the c axis? Can the experiment be repeated for trigonal Se?

[Reply3-3]

We are also really interested in these points. Below is our current expectation.

If the current is applied not along the c axis, but along, for example, the a axis (*i.e.*, the x axis), the current-induced spin polarization would be smaller than that in the present experiment. This is because the spin of the uppermost valence band, $\mathbf{s}(\mathbf{k}) = (\sim 0, \sim 0, 3Sk_z / 2\sqrt{S^2k_z^2 + \Delta^2})$, has only the z -axis component proportional to k_z near the point H (H'). As a result, the current-induced imbalance between the populations of the holes with $k_x > 0$ and with $k_x < 0$ does not yield a net spin polarization. However, the effect of the tiny third-order term ($\propto k_x(k_x^2 - 3k_y^2)$) in the energy dispersion (*Braun et al. phys. stat. sol. (b)* **53**, 635 (1972)) might cause the x and y components of $\mathbf{s}(\mathbf{k})$, which would result in a detectable current-induced net spin polarization. We appreciate this comment, and we agree that the current-direction, the temperature and the material dependences of the present current-induced magnetization should be clarified experimentally. However, it would take years to do them (please also see Reply 3-2); they are beyond the scope of the present study.

[Comment 3-4]

(a) Fig 1c shows separate Fermi surfaces near the H(H') points. However, Doi (Ref 17 and its companion paper in 1970 in the same journal) gives a depth of the uppermost valence band of 1.1 meV. At 100K, this feature is irrelevant.

(b) Also, the description of this panel on p. 4 is incorrect -- the conduction band has a radial spin texture, but the valence band does not. Please read Ref. 18 carefully, and/or diagonalize the Hamiltonian given by Doi and examine the expectation values of spin operators for the highest-energy eigenstate.

[Reply 3-4]

(a) As the reviewer noted, the features of the Fermi surfaces in Fig. 1c in the original manuscript were irrelevant at 100 K. We, therefore, calculated the correct distribution of the holes with a Fermi level at 100 K using the acceptor concentration of $5 \times 10^{15} \text{ cm}^{-3}$, an acceptor level of 1.4 meV, and a band dispersion of $E(\mathbf{k}) = A(k_x^2 + k_y^2) + Bk_z^2 + \sqrt{S^2k_z^2 + \Delta^2} - \Delta - E_0$, where $\mathbf{k} = (k_x, k_y, k_z)$ is a wave vector from the H or H' points, $A = -32.6 \text{ eV \AA}^2$, $B = -36.4 \text{ eV \AA}^2$, $|S| = 2.47 \text{ eV \AA}$, $\Delta = 63 \text{ meV}$, and $E_0 = 2.4 \text{ meV}$. As a result, although the Fermi surfaces collapse at 100 K, all of the excited holes were found to belong to the uppermost valence band. Since the mechanism underpinning the current-induced spin polarization does not require a Fermi degeneracy, the non-degenerate distribution does not alter our claim. In order to clarify this point, we have revised the manuscript in the ways outlined below.

We have revised Fig. 1; the revised figure contains the actual position of the Fermi level (*i.e.*, the chemical potential), the non-degenerate distribution of the holes at 100 K, and the spin textures of the uppermost valence band.

In addition, in the legend of Figure 1,

The text

“c, First Brillion zone and spin-polarized Fermi surfaces of p-type tellurium with a low hole density ($n_H < 10^{17} \text{ cm}^{-3}$). The spins on the hole pockets are almost parallel to the c axis, in radial fashion from the H (H') point.”

has been replaced by

“c, The first Brillouin zone and distribution of the holes at 100 K. Although the holes are not Fermi-degenerate at 100 K, they only belong to the uppermost valence band. The colours of the lower panels represent the distribution function of the holes at $T = 100$ K, $f(\mathbf{k}) = 1/[\exp\{(-E(\mathbf{k})+\mu(100\text{ K}))/k_B T\}+1]$ (where k_B is the Boltzmann constant), and the lines indicate constant $f(\mathbf{k})$ contours: $f(\mathbf{k}) = 0.002, 0.004, 0.006, 0.008$ and 0.01 . The arrows represent the direction and the magnitude of the spin of the electron of the uppermost valence band, $\mathbf{s}(\mathbf{k}) = (\sim 0, \sim 0, 3Sk_z/2\sqrt{S^2k_z^2 + \Delta^2})$. The spins are almost parallel to the c axis and radial-like from the H (H') point.”

And the text

“The chemical potential at 100 K is described by the broken line $\mu(T = 100\text{ K})$.”

has been added.

(b) We understand that the spin texture of the uppermost valence band near point H (H') only has a z -axis (c -axis) component, as shown in Fig. 1c, in the original manuscript. The spin texture was calculated as being $\mathbf{s}(\mathbf{k}) = (\sim 0, \sim 0, 3Sk_z/2\sqrt{S^2k_z^2 + \Delta^2})$ near the H and H' points. If the reviewer meant that the spin texture of the uppermost valence band are not radial, in the sense that the term “radial” means radial not from the H or H' points but from the centre of each Fermi pocket, then this is a simple misunderstanding. We have used “radial” in this paper to mean radial from point H (H'). Indeed, we wrote in the figure legend of Fig. 1 in the original manuscript that “The spins on the hole pockets are almost parallel to the c axis, in radial fashion from the H (H') point.” However, if the reviewer meant that the spin texture is not radial in the sense that \mathbf{k} and $\mathbf{s}(\mathbf{k})$ are not *strictly* parallel to one another, then the reviewer's comment is formally correct. Nevertheless, we used the term “radial” deliberately in the present study in order to emphasize the difference of this system from a system that has a Rashba-type circular spin texture. The system with a radial spin texture and that with a circular spin texture cause different types of current-induced spin polarizations; the former and the latter exhibit $\langle s \rangle // I_z$ and $\langle s \rangle \perp I_z$, respectively. However, thanks to this comment, we noticed that the original manuscript was unclear, and so we have revised the manuscript accordingly, as described below.

In pages 4 and 5 of the revised manuscript, the text

“The energy dispersion of the uppermost valence band near the H and H' points is well approximated by $E(\mathbf{k}) = A(k_x^2 + k_y^2) + Bk_z^2 + \sqrt{S^2k_z^2 + \Delta^2} - \Delta - E_0$, where $\mathbf{k} = (k_x, k_y, k_z)$ is a wave vector measured from the H or H' points, $A = -32.6\text{ eV \AA}^2$, $B = -36.4\text{ eV \AA}^2$, $S = \pm 2.47\text{ eV \AA}$ ($-$ for $P3_121$ and $+$ for $P3_221$), $\Delta = 63\text{ meV}$, and $E_0 = 2.4\text{ meV}$ (Refs. 18–24).”

has been added with additional references for

20. Nakao, K., Doi, T. & Kamimura, H. The Valence Band Structure of Tellurium. III. The Landau Levels. *J. Phys. Soc. Japan* **30**, 1400–1413 (1971).

21. Braun, E., Neuringer, L. J. & Landwehr, G. Valence Band Structure of Tellurium from Shubnikov-de Haas Experiments. *Phys. Status Solid (b)* **53**, 635–650 (1972).
22. Ivchenko, E. L. & Pikus, G. E. Natural optical activity of semiconductors (tellurium). *Sov. Phys. Solid State* **16**, 1261–1265 (1974).
23. Dubinskaya, L. S. & Farbshtein, I. I. Natural optical activity and features of the structure of the electronic energy spectrum of tellurium. *Sov. Phys. Solid State* **20**, 437–441 (1978).
24. Stolze, H., Lutz, M. & Grosse, P. The Optical Activity of Tellurium. *Phys. Status Solidi (b)* **82**, 457–466 (1977).

The text

“Note that the Fermi surfaces for tellurium have a radial spin texture, in contrast to Rashba-type circular texture, i.e., the spins on the K-H, K'-H', and H-H' lines are parallel to each line because of the threefold screw symmetry on the K-H (K'-H') lines and the twofold symmetry on the H-H' lines without any mirror symmetry. Indeed, a theoretical investigation¹⁸ has confirmed the presence of Fermi surfaces with a radial spin texture. ”

has been replaced by

“Note that the spin texture of tellurium tends to be radial from the H (H') point, in contrast to that of a Rashba-type circular texture, i.e., the crystal symmetry of tellurium imposes the requirement that the spins on the K-H, K'-H' and H-H' lines are parallel to each line, because of the threefold screw symmetry on the K-H (K'-H') lines and the twofold symmetry on the H-H' lines without any mirror symmetry. Indeed, theoretical investigations^{19,20,25} have confirmed that the conduction bands have simple radial spin textures, and that the uppermost valence band also has a radial-like, but almost c-axis oriented, spin texture of the electron,

$$\mathbf{s}(\mathbf{k}) = (\sim 0, \sim 0, 3Sk_z / 2\sqrt{S^2k_z^2 + \Delta^2}) \text{ near the H(H') point. Note that the spin textures for right- and left-handed crystals are opposite to one another, because the two crystal structures are related by the spatial inversion.}”$$

[Comment 3-5]

No comparison to numerical estimates from theory is presented here. In other words, does 1G shift in ~ 100 A/cm² make sense? This will require some knowledge of the Knight shift but that can presumably come from temperature dependence of carrier spin polarization in the absence of a current. It also requires knowledge of Fermi surface shift but that is easily obtainable in linear response from the conductivity and effective mass (to determine scattering time). Follow a procedure similar to that given in Phys. Rev. B 93, 220404(R) (2016), using the appropriate dispersion expansion.

[Reply 3-5]

(This Reply is almost the same to Reply 2-2 & 2-3, because the Comment 2-2 & 2-3 and Comment 3-5 are essentially the same.)

As the reviewer commented, we appreciate the importance of (1) estimation of the amount of the current-induced magnetization from the current-induced NMR shift observed and the hyperfine coupling and (2) comparison of the estimated magnetization with the result of the theoretical calculation that the reviewer #3 suggested. However, at present, it is impossible to estimate the amount of the induced magnetization because there is no reliable source of information on the hyperfine coupling in elemental trigonal tellurium. In other words, reliable K (Knight shift)– χ (spin susceptibility) analyses have not been achieved, despite the long length of time of the elemental tellurium study. This is because elemental trigonal tellurium is a semiconductor with a band gap E_g of ~ 330 meV, which is much larger than room temperature, and thus the spin susceptibility is too small to be detected. Indeed, the total magnetic susceptibility does not show a temperature dependence in non-doped pure crystals below room temperature (Fukuroi *et al. J. Jap. Inst. Metals* **19**, 118–122 (1955) and Fisher *et al. J. Phys. Chem. Solids* **17**, 246–253 (1961)); the experimental value of the spin susceptibility is, therefore, not available.

Nevertheless, we will provide a rough and tentative estimate of the current-induced magnetization. In Ref. 27 (Selbach, H., Kanert, O. & Wolf, D. *Phys. Rev. B* **19**, 4435–4443 (1979)), the Knight shift data of the intrinsic region (350–700 K) is reported. We estimated the hyperfine coupling to be 5.4×10^3 T/ μ_B (where μ_B is the Bohr magneton) by using the reported Knight shift data and theoretically calculated spin susceptibility; the details are provided below.

We calculate the high-temperature spin susceptibility under a magnetic field along the c axis, according to the procedure discussed in Ref. 27. First, we adopt the approximate form of the spin susceptibility of trigonal tellurium caused by non-degenerate electrons and holes thermally excited across the band gap: $\chi_s = [\mu_B^2 n_e(T) + (2\mu_B)^2 n_h(T)] / k_B T$, where $n_e(T)$ and $n_h(T)$ are the density of the electrons and holes, respectively. Next, we assume the density of the electrons and holes to be $n_e(T) = n_h(T) = (2/h^3)(2\pi k_B T)^{3/2} (m_e^* m_h^*)^{3/4} e^{-E_g/2k_B T}$, where h is the Planck constant, and m_e^* and m_h^* are the density-of-states effective mass of the conduction and valence bands, respectively. By substituting $m_e^* = 0.091m_0$ and $m_h^* = 0.143m_0$ (where m_0 is the free electron mass) (Shinno *et al. J. Phys. Soc. Japan* **35**, 525–533 (1973)), we obtain $\chi_s = \mu_B \times [7.50 \times 10^{14} (\text{cm}^{-3} \text{T}^{-1} \text{K}^{-1/2})] \times [T(\text{K})]^{1/2} \times \exp(-E_g / 2k_B T)$. Lastly, we compare this susceptibility with the reported Knight shift, $K = [1.38 \times 10^{-4} (\text{K}^{-1/2})] \times [T(\text{K})]^{1/2} \times \exp(-E_g / 2k_B T)$; this allows us to obtain a hyperfine coupling of 5.4×10^3 T/ μ_B . We note that this value does not represent the hyperfine coupling of the uppermost valence band, but rather it is the average value of those of the uppermost valence band and the conduction bands, which can be different. In addition, we neglect the effect of the temperature dependence of the orbital magnetism, which can also contribute to the temperature dependence of the Knight shift. As a result, the estimated hyperfine coupling is not very reliable.

Nevertheless, if we adopt this hyperfine coupling value (*i.e.*, 5.4×10^3 T/ μ_B), the spectral shift of ~ 0.7 Gauss (under 82 A cm^{-2}) observed in the present study yields a magnetization of $1.3 \times 10^{-8} \mu_B$ per site.

Next, we try to compare the above rough estimation of the current-induced magnetization with a theoretical calculation. When an electric field is applied along the c (z) axis, a current-induced spin polarization $\langle s_z \rangle$ and an electric current density $\langle j_z \rangle$ can be expressed by a Boltzmann transport equation approach in the relaxation-time approximation (Li *et al. Phys. Rev. B* **93**, 220404 (2016)):

$$\langle s_z \rangle = \frac{3}{2} V \int_{BZ} \frac{d\mathbf{k}}{(2\pi)^3} s_z^h(\mathbf{k}) v_z^h(\mathbf{k}) e E_z \tau \left(\frac{\partial f_0^h}{\partial \epsilon} \right)_{\epsilon = E^h(\mathbf{k})},$$

$$\langle j_z \rangle = -e \int_{BZ} \frac{d\mathbf{k}}{(2\pi)^3} (v_z^h(\mathbf{k}))^2 eE\tau \left(\frac{\partial f_0^h}{\partial \mathcal{E}} \right)_{\mathcal{E}=E^h(\mathbf{k})},$$

where V is a system volume, e (>0) is the elementary charge, \mathbf{k} is the wave vector of a hole, $s_z^h(\mathbf{k})$ is the z -axis component of the spin of a hole at \mathbf{k} , $v_z^h(\mathbf{k})$ is the z -axis component of the group velocity of a hole at \mathbf{k} , E_z is the z -axis component of an electric field, τ is the scattering time, $E^h(\mathbf{k})$ is the energy dispersion of a hole, $(\partial f_0^h / \partial \mathcal{E})_{\mathcal{E}=E^h(\mathbf{k})}$ is a derivative of the equilibrium distribution function of the holes at $E^h(\mathbf{k})$ with respect to an energy, and BZ denotes the first Brillouin zone. If the energy dependence of τ is neglected, then the following equation is obtained:

$$\langle s_z \rangle = -\frac{3}{2} V \frac{\int_{BZ} d\mathbf{k} s_z^h(\mathbf{k}) v_z^h(\mathbf{k}) \left(\frac{\partial f_0^h}{\partial \mathcal{E}} \right)_{\mathcal{E}=E^h(\mathbf{k})}}{e \int_{BZ} d\mathbf{k} (v_z^h(\mathbf{k}))^2 \left(\frac{\partial f_0^h}{\partial \mathcal{E}} \right)_{\mathcal{E}=E^h(\mathbf{k})}} \langle j_z \rangle.$$

Note that the value of $eE\tau$ is absent in this equation. Below, we use the actual chemical potential at 100 K, $\langle j_z \rangle = 82 \text{ A cm}^{-2}$, $E^h(\mathbf{k}) = -E(-\mathbf{k}) = -[A(k_x^2 + k_y^2) + Bk_z^2 + \sqrt{S^2 k_z^2 + \Delta^2} - \Delta - E_0]$, $s_z^h(\mathbf{k}) = -s_z(-\mathbf{k}) = (3/2) \times S k_z / \sqrt{S^2 k_z^2 + \Delta^2}$ and $v_z^h(\mathbf{k}) = -(2Bk_z + S^2 k_z / \sqrt{S^2 k_z^2 + \Delta^2}) / \hbar$, where $\mathbf{k} = (k_x, k_y, k_z)$ is a wave vector from the H or H' points, $A = -32.6 \text{ eV \AA}^2$, $B = -36.4 \text{ eV \AA}^2$, $S = -2.47 \text{ eV \AA}$ (for $P3_121$), $\Delta = 63 \text{ meV}$, and $E_0 = 2.4 \text{ meV}$. As a result, the density of the current-induced spin polarization is calculated to be $\langle s_z \rangle / V = -4.1 \times 10^{13} \text{ cm}^{-3}$. The current-induced magnetization per tellurium atom M_{atom} is calculated to be $M_{\text{atom}} = g_{J=3/2} \mu_B \langle s_z \rangle \times V_{\text{atom}} \sim 1.9 \times 10^{-9} \mu_B$, where $g_{J=3/2} = -4/3$ is the Landé g -factor of $J=3/2$ ($S=1/2$, $L=1$), and $V_{\text{atom}} = 34 \text{ \AA}^3$ is the atomic volume of tellurium. This theoretically calculated result, $\sim 10^{-9} \mu_B$ per site, is comparable to the above estimation obtained from the NMR shift, even though both the estimations are rough.

In spite of the rough agreement between the experimental estimation and the theoretical calculation, we emphasize again that the hyperfine coupling value described above is not very reliable and may contain considerable uncertainties. Thus, we do not believe that the above unreliable estimation should be included in the main manuscript. Instead, we show the above discussion in the Supplementary Information in the revised manuscript. We believe that the most important point of the present study is a qualitative one, i.e., direct proof of current-induced magnetization in a genuine bulk material is provided. The further quantitative estimation is beyond the scope of the present study.

We have revised the manuscript as follows.

The text

“(We provide a rough estimate of the present current-induced magnetization in Supplementary Information.)”

has been added in page 8 in the revised manuscript.

We have also added a new section into the Supplementary Materials in order to explain all of the above discussions, and we have added the following references:

- S1. Fukuroi, T. & Yasuhara, K. On the Magnetic Susceptibility of Tellurium and the Magnetic Anisotropy of its Single Crystal. *J. Jap. Inst. Metals* **19**, 118–122 (1955).
- S2. Fisher, G. & Hedgcock, F. T. Magnetic susceptibility and galvanomagnetic effects in pure and P-type tellurium. *J. Phys. Chern. Solids* **17**, 246–253 (1961).
- S3. Shinno, H., Yoshizaki, R., Tanaka, S., Doi, T. & Kamimura, H. Conduction Band Structure of Tellurium. *J. Phys. Soc. Japan* **35**, 525–533 (1973).
- S4. Li, P. & Appelbaum, I. Interpreting current-induced spin polarization in topological insulator surface states. *Phys. Rev. B* **93**, 220404 (2016).

[Comment 3-6]

Text from line 137-140 is nonsensical to me.

[Reply 3-6]

In lines 136–140 in the original manuscript, we wrote

“This is because a microscopic electrical current flows helically through each tellurium chain, which can be regarded as the condensed matter analogue of a solenoid. Therefore, the present current-induced magnetization may have not only a spin contribution but also an orbital contribution.”

The first sentence is the description of a current-induced orbital (helical motion) magnetization discussed in Yoda *et al. Sci. Rep.* **5**, 12024 (2015). The second sentence suggests the possibility of the orbital (helical motion) contribution in the present result. Although at present we do not know whether their work is nonsensical or not, we think that we should present various possibilities to the readers of the present paper, at least present the possible explanations provided by published papers. Nevertheless, in order to explain current-induced orbital (helical motion) magnetism more clearly, we have elected to revise the manuscript in the following ways.

The text of page 8 of the original manuscript

“...trigonal tellurium can also exhibit current-induced orbital magnetization when an electrical current is applied parallel to the helical chain²¹. This is because a microscopic electrical current flows helically through each tellurium chain, which can be regarded as the condensed matter analogue of a solenoid. Therefore, the present current-induced magnetization may have not only a spin contribution but also an orbital contribution.”

has been replaced by

“...trigonal tellurium can also exhibit current-induced orbital (helical motion of holes) magnetization when an electrical current is applied parallel to the helical chain²⁸. This is because an electric current along the helical axis causes the helical motion of a wave packet, which can be regarded as being the condensed matter analogue of a solenoid. Therefore, the present current-induced electronic magnetization may possess not only a spin contribution, but also a helical motion contribution.”

The text

“The mutual enhancement of current-induced electronic spin and orbital magnetization is an interesting future issue.”

has been replaced by

“The mutual enhancement of the current-induced electronic spin and helical motion magnetization is an interesting future issue.”

and the text

“Note that in a system with strong spin-orbit coupling, neither the spin nor orbital angular momentum is a good quantum number. Thus, in principle, the current-induced orbital magnetization and current-induced spin magnetization cannot be separated.”

has been removed.

Reviewers' comments:

Reviewer #1 (Remarks to the Author):

The authors have significantly improved their manuscript. Now all the figures and list of references are appropriate. But I am not satisfied by the present version of the text (comments are listed below).

1. The authors experimentally investigate current-induced electronic magnetization in bulk tellurium crystal using nuclear magnetic resonance (NMR) technique. But there is no reason to say that author's "finding provides a **new** stage of magnetoelectricity in bulk matter" and "opens a **new** area of magnetic field generation" (Lines 22 and 26 in the revised version of the paper). The word "new" is not appropriate here. It should be emphasized that the phenomenon of current-induced electronic magnetization (spin polarization) in bulk tellurium was experimentally observed for the first time by means of optical measurements in 1979 (Ref. 26). Then in Ref. 8 the density of current-induced spin polarization was determined as a function of applied electric current. In the paper under review, the authors carry out similar evaluation (see Supplementary Information) and their result for current density of 82 A/cm² coincides with the result of Ref. 26 with accuracy of 20%. Unfortunately, the authors do not make this comparison.

To my opinion, the main achievement of the authors is the **new approach** to study current-induced electronic magnetization in bulk crystals, namely implementation of NMR technique. It seems reasonable to describe this new approach not before but after paragraphs devoted to Refs. 8 and 26.

2. The sentence "It has been reported that applying an electric current causes modulation of the optical activity inherent to the chiral structure of trigonal tellurium (Refs. 8, 26)" is misleading (Lines 83 – 85). The effect of optical activity is caused by phenomenon of spatial dispersion (and, consequently, by a linear-in-wavevector contribution to dielectric permittivity). Papers [8, 26] deal with effect of "current-induced optical activity" which is caused by linear-in-current contribution to dielectric permittivity and is not connected with spatial dispersion. Moreover, there are crystals which can demonstrate current-induced optical activity in the absence of natural optical activity (for instance, crystals with a wurtzite lattice).

In order to avoid such misunderstanding I would suggest authors to replace words "causes modulation of the optical activity" by the words "results in effect of current-induced optical activity".

3. Authors essentially extended Supplementary Information. They calculated density of current-induced spin polarization and magnetization per tellurium atom ($-4.1 \cdot 10^{13} \text{ cm}^{-3}$ and $\sim 10^{-9} \mu_{\text{B}}$, respectively, under current density of 82 A/cm²). Also they made rough estimate of current-induced magnetization based on their own measurements of the current-induced NMR shift: $1.3 \cdot 10^{-8} \mu_{\text{B}}$. But it is difficult to agree with authors that the values $10^{-9} \mu_{\text{B}}$ and $1.3 \cdot 10^{-8} \mu_{\text{B}}$ are comparable. The difference is about one order of magnitude.

On the other hand, this discrepancy may be explained by the absence of reliable data on the value of the hyperfine coupling in elemental trigonal tellurium. In principle, the authors' results give the opportunity to determine the value of the hyperfine coupling on the base of the theoretical value of the current-induced magnetization per tellurium atom of $\sim 10^{-9} \mu_{\text{B}}$ and experimental value of the current-induced NMR shift (~ 0.7 Gauss). Rather high accuracy of the theoretical model of the current-induced spin polarization in tellurium was independently confirmed by simulation of the current-induced optical activity (without any fitting parameter) and its comparison with the experimental data (Ref. 8).

Because now there is no independent reliable data on the hyperfine coupling in elemental trigonal tellurium, one can not claim that current-induced NMR shift provides more **direct** access to current-induced electronic magnetization than measurements of the current-induced optical activity. It

is related to Lines 18, 85, 88.

4. In Lines 46-49 it is desirable to replace the words "non-centrosymmetric materials" by the words "gyrotropic materials".

5. The authors significantly improved the panels (b) and (c) in Figure 1. After this modification of the figure, corresponding modification of the text should be performed. In particular, the sentences

"Tellurium has spin-polarized Fermi surfaces as a result of its strong spin-orbit interaction and inversion asymmetry. Note that the spin texture of tellurium tends to be radial from the H (H') point..." (Lines 70 -72 in the revised version of the paper)

may be replaced by the following:

"Near the top of the uppermost valence band, tellurium has spin-polarized isoenergetic surfaces as a result of strong spin-orbit interaction and inversion asymmetry. Note that the spin texture of tellurium tends to be uniaxial from the H (H') point..."

Reviewer #2 (Remarks to the Author):

I believe that the authors have adequately addressed the points of concern from my previous report. In my opinion, the article, as written, is appropriate for publication in Nature Communications.

Reviewer #3 (Remarks to the Author):

I still have many doubts about this paper, but I think it should be published so that the community can debate it.

Replies to Reviewer #1's comments

We really thank Reviewer #1 for taking the time to carefully review our manuscript and provide useful comments on it. In particular, his/her deep expertise about the optical studies of tellurium and the symmetry of the current-induced magnetization has helped us to improve the quality of our manuscript. Point-by-point replies are provided below.

[Comment 1]

1. The authors experimentally investigate current-induced electronic magnetization in bulk tellurium crystal using nuclear magnetic resonance (NMR) technique. But there is no reason to say that author's "finding provides a **new** stage of magnetoelectricity in bulk matter" and "opens a **new** area of magnetic field generation" (Lines 22 and 26 in the revised version of the paper). The word "new" is not appropriate here. It should be emphasized that the phenomenon of current-induced electronic magnetization (spin polarization) in bulk tellurium was experimentally observed for the first time by means of optical measurements in 1979 (Ref. 26). Then in Ref. 8 the density of current-induced spin polarization was determined as a function of applied electric current. In the paper under review, the authors carry out similar evaluation (see Supplementary Information) and their result for current density of 82 A/cm² coincides with the result of Ref. 26 with accuracy of 20%. Unfortunately, the authors do not make this comparison. To my opinion, the main achievement of the authors is the **new approach** to study current-induced electronic magnetization in bulk crystals, namely implementation of NMR technique. It seems reasonable to describe this new approach not before but after paragraphs devoted to Refs. 8 and 26.

[Reply1]

First of all, we greatly appreciate the reviewer's comment, which helped us to introduce the present study more appropriately. As the reviewer commented, the observation of the current-induced optical activity in 1979 (Ref. 19 = Ref. 26 in the previous manuscript) is the first experimental result which implied the current-induced spin polarization in bulk tellurium (although the work did not explicitly mention the current-induced spin polarization). And also, the recent optical measurement with a theoretical calculation (Ref. 8) discusses the quantitative relation between the current-induced optical activity and the current-induced spin polarization. According to the reviewer's comment, we revised the manuscript to emphasize these pioneering studies before introducing our approach and to explicitly refer them as the first experimental study which discussed the current-induced magnetization in tellurium. Nevertheless, there are two points that we do not agree with in the reviewer's comment.

(1) We think that the word "new" in Lines 22 and 26 are still appropriate. Although the current-induced magnetization in tellurium was first implied by means of optical measurements at 1979, this effect has not been treated or pursued in the context of a bulk phenomenon in spite of its great potential. Indeed, the current-induced bulk magnetization has not been recognized widely and has been discussed only in the two works Refs. 8 and 19 on bulk tellurium, although there are many bulk systems with spin-split energy bands, which have recently been attracted much attention, for example the bulk Rashba system BiTeI. Thus, considering the recent development or "revival" of the physics of the spin-split bands, the current-induced bulk magnetization in tellurium should be emphasized as "*a new stage of magnetoelectricity in bulk matter*". In addition, to the best of our knowledge, our work is the first time that the current-induced electronic magnetization in tellurium is connected to a new method to generate a magnetic field beyond Ampere's law. Thus, the two phrases containing the word "new" pointed by the reviewer should remain unchanged.

(2) We think that the present NMR studies provide not only an alternative approach to detect the current-induced electronic magnetization but also more direct detection of it than the optical studies (see also Reply 3). In Comment 3, the reviewer commented that the current-induced NMR shift is not a direct probe compared with the current-induced optical activity. However, we stress that the current-induced NMR shift itself indicates the generation of local electronic magnetization *without any theoretical assumption or evaluation of the current-induced electronic magnetization*. In contrast, the optical studies require separating the “pure” current-induced optical activity from the natural optical activity inherent in the structure of tellurium via careful discussions, and connecting the “pure” current-induced optical activities to the magnetization by calculating a microscopic model of tellurium. Indeed, Ref. 8 is devoted to these careful discussions and calculations. We really admire Ref. 8 for the depthful discussions and do not doubt that the optical study is the first one that captured the sign of the current-induced magnetizations, through convincing discussions. The point is that our work provides immediately obvious proof of the current induced-magnetization which does not need a complicated discussion. We believe, therefore, it is not overstatement that the present results provide the “direct” observation of the current-induced bulk electronic magnetization in tellurium.

According to the reviewer’s suggestion, we have compared the result of our calculation of the current-induced spin polarization with the result in Ref. 8. (We guess that the reviewer probably intended to refer not Ref. 19 (=Ref. 26 in the previous manuscript) but Ref. 8 for this comparison.) The reviewer commented that the two results coincide with accuracy of 20%; we found better agreement (with accuracy less than a few per cents), considering the different conditions between the two studies, such as temperature, carrier densities, and band parameters.

The manuscript has been revised as follows.

In page 1 of the original manuscript, the text

“Here we show a new class of bulk magnetoelectric effect, by revisiting elemental trigonal tellurium.”

has been replaced by the text

“Here we show the direct observation of a new class of bulk magnetoelectric effect, by revisiting elemental trigonal tellurium.”.

In page 3 of the original manuscript, the text

“What we demonstrate in the present work is that this phenomenon can be extended to bulk physics. (In this study, the term “bulk” refers to an infinite system free of boundary effects including the local strain caused by a heterointerface.) We demonstrate that an applied current can induce bulk electronic magnetization in non-centrosymmetric materials, because the inversion asymmetry in bulk crystals produces spin-split bulk bands. We choose a bulk crystal of a chiral semiconductor, elemental tellurium, as a playground, and show direct evidence for current-induced electronic magnetization using nuclear magnetic resonance (NMR) measurements.”

has been replaced by the text

“(In this study, the term “bulk” refers to an infinite system free of boundary effects including the local strain caused by a heterointerface.) To extend this phenomenon to bulk physics is an intriguing issue. An applied current can induce bulk electronic magnetization in non-centrosymmetric materials (to be precise “gyrotropic”¹⁸ non-centrosymmetric materials), because the inversion asymmetry in bulk crystals produces spin-split bulk bands. Elemental tellurium, which is a bulk crystal of a chiral semiconductor composed of heavy atoms, is an ideal playground for this issue. Indeed, it has been reported that applying an electric current to bulk tellurium causes the current induced optical activity^{8,19}. This is the first experimental result that captured a sign of the current-induced electronic magnetization in bulk tellurium¹⁹, although careful discussions, such as about separation of this additional optical activity from the inherent natural optical activity, are needed. To firmly establish the current-induced bulk electronic magnetization in tellurium, it is required to detect it simply and more directly by a probe capable of direct, sensitive, and microscopic detection of local electronic magnetization. In this paper, we demonstrate the direct observation of current-induced electronic magnetization using nuclear magnetic resonance (NMR) measurements.”

with an additional reference for

18. Ganichev, S. D. & Golub, L. E. Interplay of Rashba/Dresselhaus spin splittings probed by photogalvanic spectroscopy -A review. *Phys. Status Solidi (b)* **251**, 1801–1823 (2014).

In page 5 of the original manuscript, the text

“It has been reported that applying an electric current causes modulation of the optical activity inherent to the chiral structure of trigonal tellurium^{8,26}. This may be an indirect sign of current-induced electronic magnetization, although this remains a matter of debate. Our study provides definite evidence for current-induced electronic magnetization since NMR is capable of direct, sensitive, and microscopic detection of local electronic magnetization.”

has been removed.

The text

“Note that the present calculation quantitatively reproduces the result of a similar calculation in Ref. 8 with accuracy less than a few per cents, considering the different conditions between the two studies, such as temperature, carrier densities, and band parameters.”

has been added into page 3 of the revised Supplementary Information.

[Comment 2]

The sentence “It has been reported that applying an electric current causes modulation of the optical activity inherent to the chiral structure of trigonal tellurium (Refs. 8, 26)” is misleading (Lines 83 – 85). The effect of optical activity is caused by phenomenon of spatial dispersion (and, consequently, by a linear-in-wavevector contribution to dielectric permittivity). Papers [8, 26] deal with effect of “current-induced optical activity” which is caused by linear-in-current contribution to dielectric permittivity and is not connected with spatial dispersion. Moreover, there are crystals which can demonstrate current-induced optical activity in the absence of natural optical activity (for instance, crystals with a wurtzite lattice).

In order to avoid such misunderstanding I would suggest authors to replace words “causes modulation of the optical activity” by the words “results in effect of current-induced optical activity”.

[Reply 2]

Thanks to this comment, we noticed that the original description was misleading. Accordingly, we have revised the manuscript as follows.

In Lines 83 – 85 of the original manuscript, the text

“It has been reported that applying an electric current causes modulation of the optical activity inherent to the chiral structure of trigonal tellurium^{8,26}.”

has been removed.

In page 3 in the revised manuscript, the text

“...it has been reported that applying an electric current to bulk tellurium causes current induced optical activity^{8,19}.”

has been added.

[Comment 3]

3. Authors essentially extended Supplementary Information. They calculated density of current-induced spin polarization and magnetization per tellurium atom ($-4.1 \cdot 10^{13} \text{ cm}^{-3}$ and $\sim 10^{-9} \mu_B$, respectively, under current density of 82 A/cm²). Also they made rough estimate of current-induced magnetization based on their own measurements of the current-induced NMR shift: $1.3 \cdot 10^{-8} \mu_B$. But it is difficult to agree with authors that the values $10^{-9} \mu_B$ and $1.3 \cdot 10^{-8} \mu_B$ are comparable. The difference is about one order of magnitude.

On the other hand, this discrepancy may be explained by the absence of reliable data on the value of the hyperfine coupling in elemental trigonal tellurium. In principle, the authors' results give the opportunity to determine the value of the hyperfine coupling on the base of the theoretical value of the current-induced magnetization per tellurium atom of $\sim 10^{-9} \mu_B$ and experimental value of the current-induced NMR shift (~ 0.7 Gauss). Rather high accuracy of the theoretical model of the current-induced spin polarization in tellurium was independently confirmed by simulation of the current-induced optical activity (without any fitting parameter) and its comparison with the experimental data (Ref. 8).

Because now there is no independent reliable data on the hyperfine coupling in elemental trigonal tellurium, one can not claim that current-induced NMR shift provides more **direct** access to current-induced electronic magnetization than measurements of the current-induced optical activity. It is related to Lines 18, 85, 88.

[Reply 3]

As the reviewer commented, it is probable that the discrepancy between the evaluations of the current-induced spin polarization from the experimental result and from the theoretical calculation is due to the absence of the reliable hyperfine coupling data. Determination of the hyperfine coupling from the theoretical estimation and the current-induced NMR shift is not the goal of this research.

The main point of the reviewer's comment is that the current-induced NMR shift is not a probe that more directly detects the current-induced magnetization than the optical activity. However, we do not think so. The absence of the reliable data on the hyperfine coupling constant does not spoil the merit of the NMR measurement. We stress that the NMR shift itself indicates the generation of local electronic magnetization without any theoretical assumption or evaluation of the current-induced electronic magnetization. As explained in Reply 1, we believe, it is not overstatement that the present NMR study provide the "direct" observation of the current-induced electronic magnetization.

[Comment 4]

In Lines 46-49 it is desirable to replace the words “non-centrosymmetric materials” by the words “gyrotropic materials”.

[Reply 4]

After careful reading this Comment 4 and the Comment 1-2 in the previous review, we understand what the reviewer really means. We thought that “gyrotropic materials” in the comments meant materials that exhibit natural optical activity. However, the reviewer probably uses the extended and sophisticated definition of the word “gyrotropic” [Ganichev *et al.*, *Phys. status solidi (b)* **251**, 1801–1823 (2014)]: gyrotropic crystals are crystals in which certain components of the polar and axial vectors are transformed in the same manner for all symmetry transformations of the crystals. Assuming that, we basically agree with the reviewer’s comment. Thanks to the reviewer's deep insight into the symmetry of current-induced magnetization, we are happy to revise the manuscript as follows.

In the revised manuscript, the text

“(to be precise, “gyrotropic”¹⁸ non-centrosymmetric materials)”

has been added into page 3 of the revised manuscript with an additional reference for

18. Ganichev, S. D. & Golub, L. E. Interplay of Rashba/Dresselhaus spin splittings probed by photogalvanic spectroscopy -A review. *Phys. Status Solidi (b)* **251**, 1801–1823 (2014).

[Comment 5]

The authors significantly improved the panels (b) and (c) in Figure 1. After this modification of the figure, corresponding modification of the text should be performed. In particular, the sentences

“Tellurium has spin-polarized Fermi surfaces as a result of its strong spin-orbit interaction and inversion asymmetry. Note that the spin texture of tellurium tends to be radial from the H (H') point...” (Lines 70 -72 in the revised version of the paper)

may be replaced by the following:

“Near the top of the uppermost valence band, tellurium has spin-polarized isoenergetic surfaces as a result of strong spin-orbit interaction and inversion asymmetry. Note that the spin texture of tellurium tends to be uniaxial from the H (H') point...”

[Reply 5]

We appreciate the reviewer's careful reading of our manuscript and useful suggestions. Thanks to this comment we noticed that the original sentence was unclear. In the former sentence of the original manuscript, “*Tellurium has spin-polarized Fermi surfaces as a result of its strong spin-orbit interaction and inversion asymmetry.*”, we intended to refer not only to the uppermost valence bands but also to other energy bands including the conduction bands. Thus, we simply used ‘isoenergetic surfaces’ instead of “Fermi surfaces” in the revised manuscript, as the reviewer suggested. We still think that the word “radial” in the latter sentence is appropriate. Indeed, the spin texture of the uppermost valence band near the H (H') point is not radial in the sense that \mathbf{k} and $\mathbf{s}(\mathbf{k})$ are not *strictly* parallel to one another. Nevertheless, we used the term “radial” deliberately in the present study in order to emphasize the difference between this system and a system that has a Rashba-type circular spin texture. The systems with a radial spin texture and a circular spin texture cause different types of current-induced spin polarizations; the former and the latter tend to exhibit $\langle s \rangle // I$ and $\langle s \rangle \perp I$, respectively (although, to be exact, the current-induced spin polarization should be quite anisotropic in the tellurium, as discussed in the previous replies). To provide more detailed information of the spin texture of the uppermost valence bands, we wrote down the form of the spin texture $\mathbf{s}(\mathbf{k})$ in page 5, and stated “... the uppermost valence band also has a radial-like, but almost **c-axis oriented**, spin texture of the electron, ...” in page 5. Thus, the “uniaxial” properties of the spin texture have already been described in the original manuscript. Accordingly, we revised the manuscript as follows.

In page 5 of the original manuscript, the text

“*Tellurium has spin-polarized Fermi surfaces as a result of its strong spin-orbit interaction and inversion asymmetry.*”

has been replaced by the text

“*Tellurium has spin-polarized isoenergetic surfaces as a result of its strong spin-orbit interaction and inversion asymmetry.*” .